# Slot State Space Models

**Jindong Jiang**[*]
Rutgers University

**Fei Deng**
Rutgers University

**Gautam Singh**
Rutgers University

**Minseung Lee**
KAIST

**Sungjin Ahn**[*]
KAIST

## Abstract

Recent State Space Models (SSMs) such as S4, S5, and Mamba have shown remarkable computational benefits in long-range temporal dependency modeling. However, in many sequence modeling problems, the underlying process is inherently modular and it is of interest to have inductive biases that mimic this modular structure. In this paper, we introduce SlotSSMs, a novel framework for incorporating independent mechanisms into SSMs to preserve or encourage separation of information. Unlike conventional SSMs that maintain a monolithic state vector, SlotSSMs maintains the state as a collection of multiple vectors called slots. Crucially, the state transitions are performed independently per slot with sparse interactions across slots implemented via the bottleneck of self-attention. In experiments, we evaluate our model in object-centric learning, 3D visual reasoning, and long-context video understanding tasks, which involve modeling multiple objects and their long-range temporal dependencies. We find that our proposed design offers substantial performance gains over existing sequence modeling methods. Project page is available at `https://slotssms.github.io/`

## 1 Introduction

State space models (SSMs) have recently emerged as a promising class of sequence models, achieving remarkable success in language modeling [27, 58, 24, 50, 9] due to their long-term memory capability and computational efficiency. Compared to Transformers [4] whose attention mechanisms also facilitate capturing long-range dependencies, SSMs are more efficient during both training and inference. Notably, SSMs offer parallel training with sub-quadratic complexity, and recurrent generation with constant cost per time step. These benefits have motivated the application of SSMs to sequences of other modalities such as audio [19] and video [10].

Typically, SSMs use a monolithic state vector to summarize all past information. This design can struggle to model sequences with modular underlying structures, which are common in physical processes and real-world dynamics. For example, physical objects largely follow independent dynamics based on their own properties, with strong interactions happening only sparsely (*e.g.*, when objects come in close contact). A monolithic state vector would excessively entangle the dynamics of different entities, thereby hurting generalization. It could be beneficial to incorporate inductive biases for independent mechanisms [20] into the sequence modeling architecture.

Recent progress in object-centric learning [46, 54, 36] has led to several methods for discovering modular object-centric structures and modeling their dynamics from videos with no or only weak supervision [39, 13, 57]. Similar to RIMs [20], they build modularity into the RNN architecture to separately keep track of the dynamics of each object. However, RNNs are prone to vanishing

---

[*]Correspondence to `jindong.jiang@rutgers.edu` and `sungjin.ahn@kaist.ac.kr`.

38th Conference on Neural Information Processing Systems (NeurIPS 2024).

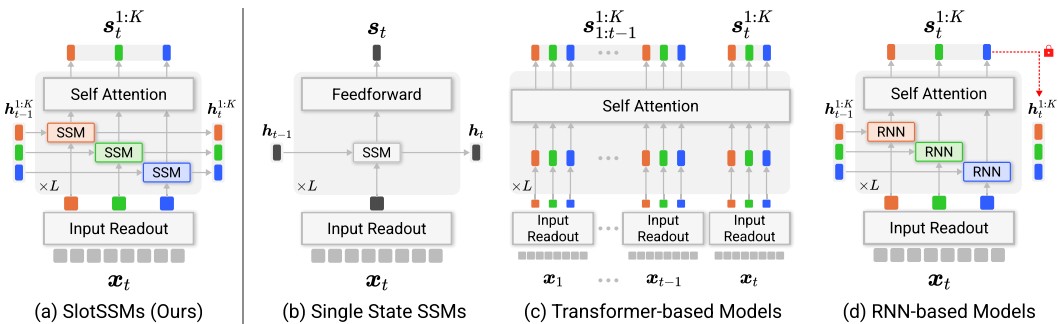

**Figure 1: SlotSSMs vs existing models.** (a) SlotSSMs incorporate modularity through independent state transitions and sparse interactions via self-attention. (b) Traditional SSMs utilize a monolithic state vector for all past information. (c) Multi-slot Transformer-based models offer modularity but with high computational complexity. (d) Multi-slot RNN-based models have modular states but can't parallelize training (red lock). SlotSSMs combine parallelizable training, memory efficiency, and modularity for efficient temporal modeling.

gradients [51] and are not amenable to parallel training, making it hard to scale these methods up to modeling long-range effects that span hundreds of time steps.

In this paper, we propose Slot State Space Models (SlotSSMs), a novel and general SSM framework that have built-in inductive biases for discovering and maintaining independent mechanisms. Instead of using monolithic state vectors, SlotSSMs maintain a set of modular slot states whose transition dynamics are designed to be largely independent, with only sparse interaction across slots introduced through the bottleneck of self-attention. The number of slots can be flexible across the layers of SlotSSMs, allowing slots to have a different level of abstraction at each layer. Furthermore, SlotSSMs inherit the strengths of SSMs, namely parallelizable training, memory efficiency, and long-range reasoning capabilities, giving it an advantage over methods based on RNNs and Transformers.

Our contributions are summarized as follows. First, we propose SlotSSMs, a novel and general architecture that incorporates independent mechanisms into SSMs for modeling inherently modular physical processes. Second, we show that SlotSSMs can be specialized to solve object-centric learning tasks. It achieves comparable or better performance than existing RNN-based methods and the Transformer baseline that we develop, while being more computationallly efficient. Third, we further investigate the abilities of SlotSSMs as a general sequence modeling framework, demonstrating its advantages in video understanding and prediction, long-range reasoning, and 3D visual reasoning.

## 2 Preliminaries

A state space model (SSM) defines a mapping between an input sequence $e_{1:T} \in \mathbb{R}^{T \times D}$ and an output sequence $y_{1:T} \in \mathbb{R}^{T \times D}$ via the recurrence [28, 27, 58, 50]:

$$\begin{aligned} h_t &= \overline{A}_t h_{t-1} + \overline{B}_t e_t \,, \\ y_t &= C_t h_t \,. \end{aligned} \tag{1}$$

Here, $T$ is the sequence length; $e_t, y_t \in \mathbb{R}^D$ are input and output vectors at time $t$; and $h_t \in \mathbb{R}^H$ is the hidden state summarizing the history $e_{\leq t}$. The matrices $\overline{A}_t \in \mathbb{R}^{H \times H}$, $\overline{B}_t \in \mathbb{R}^{H \times D}$, and $C_t \in \mathbb{R}^{D \times H}$ are designed with learnable parameters in specific ways that encourage modeling long-range dependencies while maintaining computational efficiency. For example, $\overline{A}_t$ commonly takes a diagonal or block-diagonal form, with its (complex) eigenvalues distributed close to the unit circle at initialization [25, 27, 29, 26, 58, 50].

When $\overline{A}_t, \overline{B}_t, C_t$ are time-invariant (constant over $t$), the computation of $y_{1:T}$ can be parallelized, enabling efficient training. Recent works [24, 9] further show that conditioning these matrices on the input $e_t$ does not hinder training efficiency. They employ learnable functions $\overline{A} \colon \mathbb{R}^D \to \mathbb{R}^{H \times H}, \overline{B} \colon \mathbb{R}^D \to \mathbb{R}^{H \times D}, C \colon \mathbb{R}^D \to \mathbb{R}^{D \times H}$ to generate input-dependent matrices:

$$\overline{A}_t = \overline{A}(e_t) \,, \quad \overline{B}_t = \overline{B}(e_t) \,, \quad C_t = C(e_t) \,. \tag{2}$$

This allows the model to selectively emphasize or ignore information based on the input, leading to more flexible sequence modeling.

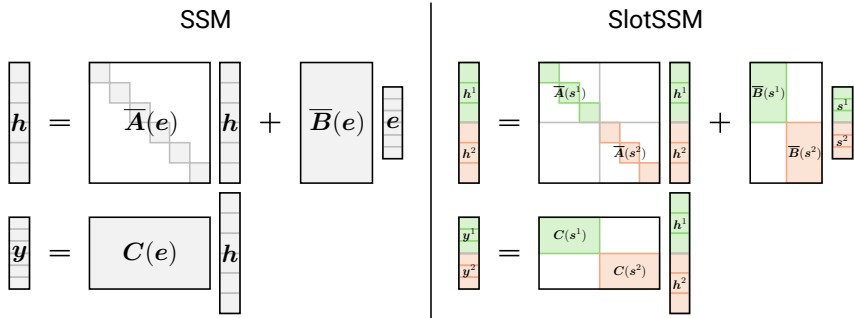

**Figure 2: SSM vs SlotSSM.** SlotSSM encourages modularity by maintaining a set of separate slot state representations, each updated independently using separate transition matrices and input matrices, allowing for more efficient and scalable modeling of complex sequences with inherent modular structures.

Due to the (block-)diagonal structure of $\overline{A}_t$ limiting cross-dimensional information flow, SSMs are typically interleaved with mixing layers (e.g., linear projections or MLPs) to mix information across dimensions. Alternatively, using dense $\overline{B}_t$ and $C_t$ matrices can also enhance mixing.

## 3 Slot State Space Models (SlotSSMs)

Standard SSMs use monolithic vectors for inputs, outputs, and hidden states, and mix information across all dimensions. This lack of modularity could cause difficulties in modeling real-world dynamics such as object interactions, where the underlying process consists of multiple entities and is inherently modular [20]. In this section, we present slot state space models (SlotSSMs), a new class of SSMs with built-in inductive biases for encouraging and preserving modularity.

Our key idea is to maintain a set of separate *slot state* representations (called slots in short), and process the slots independently and symmetrically. To do this, we format the input vector $e_t \in \mathbb{R}^D$ as a concatenation of $K$ slot representations $\{s_t^k \in \mathbb{R}^{D_s}\}_{k=1}^K$, where $D_s = D/K$. The output $y_t \in \mathbb{R}^D$ and hidden state $h_t \in \mathbb{R}^H$ are formatted similarly:

$$e_t = \texttt{concat}\left[s_t^1, \ldots, s_t^K\right], \quad y_t = \texttt{concat}\left[y_t^1, \ldots, y_t^K\right], \quad h_t = \texttt{concat}\left[h_t^1, \ldots, h_t^K\right], \quad (3)$$

where $y_t^k \in \mathbb{R}^{D_s}$ and $h_t^k \in \mathbb{R}^{H_s}$ are the output and the hidden state corresponding to slot $s_t^k$, with $H_s = H/K$. In this section, we focus on preserving modularity when the input already complies with the slot format. When coupled with a slot encoder, the SlotSSM can help encourage the emergence of modularity from unstructured inputs such as video frames, as we will discuss in Section 4.

To preserve modularity, we make sure that SlotSSM do not mix information across different slots. More precisely, the hidden state $h_t^k$ and output $y_t^k$ only integrate information from the history of the corresponding input slot $s_{\leq t}^k$. As illustrated in Figure 2 (Right), this can be achieved by making $\overline{A}_t, \overline{B}_t, C_t$ block-diagonal, where the $k$-th block is only conditioned on the $k$-th slot:

$$\overline{A}_t = \text{diag}\left(\{\overline{A}(s_t^k)\}_{k=1}^K\right), \quad \overline{B}_t = \text{diag}\left(\{\overline{B}(s_t^k)\}_{k=1}^K\right), \quad C_t = \text{diag}\left(\{C(s_t^k)\}_{k=1}^K\right). \quad (4)$$

**Implementation details.** The SlotSSM formulation in Equation 4 is general and can accommodate various choices of the $\overline{A}, \overline{B}, C$ functions. In our implementation, we adopt those from Mamba [24]. Specifically, $\overline{A}(s_t^k), \overline{B}(s_t^k), C(s_t^k)$ are themselves block-diagonal matrices with $D_s$ blocks, one for each slot dimension. The $i$-th blocks $\overline{A}^{(i)}(s_t^k) \in \mathbb{R}^{N \times N}$ and $\overline{B}^{(i)}(s_t^k) \in \mathbb{R}^{N \times 1}$ are obtained by discretizing their continuous-time counterparts $A^{(i)}$ and $B^{(i)}(s_t^k)$ using the time step $\Delta^{(i)}(s_t^k)$ and the zero-order hold (ZOH) rule:

$$\overline{A}^{(i)}(s_t^k), \overline{B}^{(i)}(s_t^k) = \text{ZOH}\left(\Delta^{(i)}(s_t^k), A^{(i)}, B^{(i)}(s_t^k)\right), \quad i = 1, \ldots, D_s. \quad (5)$$

Here, $N = H_s/D_s$ is the hidden state size per slot dimension, $A^{(i)} \in \mathbb{R}^{N \times N}$ is an input-independent learnable model parameter, and $\Delta^{(i)} : \mathbb{R}^D \to \mathbb{R}, B^{(i)} : \mathbb{R}^D \to \mathbb{R}^{N \times 1}$ are learnable functions implemented as neural networks. Similarly, the $i$-th block $C^{(i)}(s_t^k)$ is computed by the learnable function

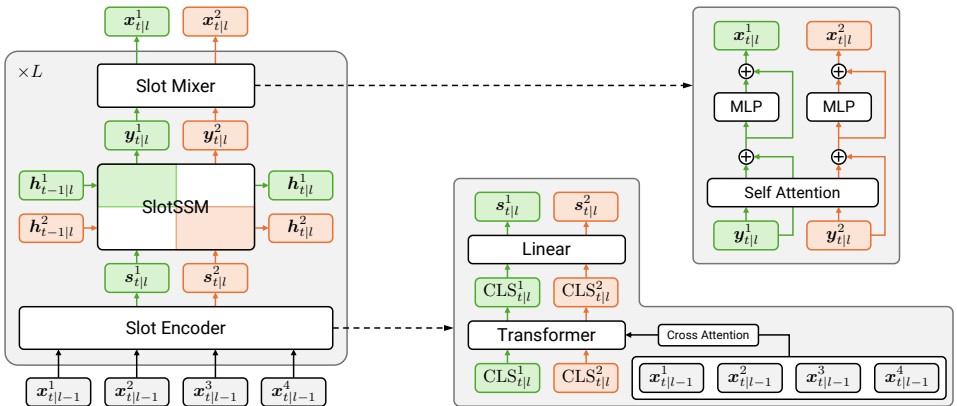

**Figure 3: Sequence modeling with SlotSSM.** Each layer includes a Slot Encoder, SlotSSM, and Slot Mixer. The Slot Encoder uses a Transformer to extract slots from inputs. The SlotSSM independently updates the slots via separate state transitions. The Slot Mixer introduces inter-slot interactions through self-attention.

$C^{(i)} \colon \mathbb{R}^D \to \mathbb{R}^{1 \times N}$. For simplicity and efficiency, $B^{(i)}$ and $C^{(i)}$ are shared across all $1 \le i \le D_s$, and $A^{(i)}$ is parameterized as a diagonal matrix.

# 4 Modular Sequence Modeling with SlotSSM

The SlotSSM proposed in Section 3 are designed to preserve modularity when the input is already separated into slots. In this section, we complement SlotSSM with a slot encoder that extracts slot representations from unstructured inputs (Section 4.1), and a slot mixer that introduces sparse interactions across slots (Section 4.2). We then present a sequence modeling architecture (Section 4.3) that encourages discovery of underlying modular processes by stacking these components.

## 4.1 Slot Encoder

We assume the unstructured input $x_t$ at each time step $t$ is represented as a sequence of $M$ tokens:

$$x_t = (x_t^1, \ldots, x_t^M), \quad x_t^m \in \mathbb{R}^{D_x} . \tag{6}$$

For example, image inputs can be CNN feature maps ($M$ is the number of cells in the feature map), or as embeddings of non-overlapping image patches ($M$ is the number of patches), as proposed in ViT [12]. To extract $K$ slot representations from $x_t$, we use $K$ learnable CLS [2] tokens $\{\mathrm{CLS}_t^k \in \mathbb{R}^{D_x}\}_{k=1}^K$ as queries and perform cross-attention with the input tokens through a Transformer [63]:

$$\{\mathrm{CLS}_t^k\}_{k=1}^K \leftarrow \mathrm{Transformer}\Big(\mathtt{q} = \{\mathrm{CLS}_t^k\}_{k=1}^K, \ \mathtt{kv} = \{x_t^m\}_{m=1}^M\Big) . \tag{7}$$

The Transformer also includes self-attention within the CLS tokens, allowing them to communicate with each other and capture information from different parts of the input, thereby facilitating the emergence of modularity. The slot representations are then obtained by applying a linear projection to the corresponding output embeddings of the CLS tokens:

$$s_t^k = \mathrm{Linear}(\mathrm{CLS}_t^k), \quad k = 1, \ldots, K . \tag{8}$$

## 4.2 Slot Mixer

The slot encoder obtains slot decomposition purely based on single time steps, which can be suboptimal. In addition, the SlotSSM processes each slot fully independently, making it hard to correct mistakenly decomposed slots or model interarctions across slots. To resolve both issues, we interleave SlotSSM with slot mixers.

---

[2]Following the tradition of ViT, we use "CLS" to distinguish learnable tokens from observation tokens.

The slot mixer consists of two residual blocks, and is applied to the outputs $\{y_t^k\}_{k=1}^K$ of the SlotSSM. The first block introduces interaction across slots through self-attention [63], whereas the second block uses MLP to further process the gathered information within each slot:

$$\left(y_t^1, \ldots, y_t^K\right) \leftarrow \left(y_t^1, \ldots, y_t^K\right) + \text{SelfAttn}\left(\text{LN}(y_t^1), \ldots, \text{LN}(y_t^K)\right) , \tag{9}$$

$$\left(y_t^1, \ldots, y_t^K\right) \leftarrow \left(y_t^1, \ldots, y_t^K\right) + \left(\text{MLP}(\text{LN}(y_t^1)), \ldots, \text{MLP}(\text{LN}(y_t^K))\right) . \tag{10}$$

Here, $\text{LN}(\cdot)$ denotes layer normalization [2]. Because $y_t^k$ carries information from the entire history of each slot, it provides the opportunity to refine the slot representations based on temporal dynamics.

## 4.3   Sequence Modeling Architecture

We now present a generic architecture for modeling sequences with modular underlying processes. Given a sequence of unstructured inputs $x_{1:T}$, our goal is to obtain a set of $K_l$ modular representations at each time step $t$ and at each layer $l$ that summarizes all underlying processes up to time $t$.

In general, the number of slots $K_l$ at each layer can be different, potentially allowing fewer but more abstract slots at higher layers. To accommodate this, we insert a slot encoder wherever the number of slots changes, and repurpose it to extract a different number of slots from existing slot representations. This is achieved by treating the slots output from the previous layer as keys and values in Equation 7. When the number of slots does not change, we can simply copy the slots from the previous layer.

As shown in Figure 3, our proposed architecture stacks the (optional) slot encoder, SlotSSM, and slot mixer together at each layer. Variables at layer $l$ are denoted with the subscript '$|l$'. The slot mixer's output from layer $l-1$, $\{x_{t|l-1}^k\}_{k=1}^{K_{l-1}}$, serves as input to layer $l$. The initial input is $\{x_{t|0}^k\}_{k=1}^{K_0}$, where $K_0 := M$. The computations at each layer $l = 1, \ldots, L$ are:

$$\{s_{t|l}^k\}_{k=1}^{K_l} = \text{SlotEncoder}\left(\{x_{t|l-1}^k\}_{k=1}^{K_{l-1}}\right) , \tag{11}$$

$$\{y_{t|l}^k\}_{k=1}^{K_l}, \{h_{t|l}^k\}_{k=1}^{K_l} = \text{SlotSSM}\left(\{s_{t|l}^k\}_{k=1}^{K_l}, \{h_{t-1|l}^k\}_{k=1}^{K_l}\right) , \tag{12}$$

$$\{x_{t|l}^k\}_{k=1}^{K_l} = \text{SlotMixer}\left(\{y_{t|l}^k\}_{k=1}^{K_l}\right) . \tag{13}$$

The final output $\{x_{t|L}^k\}_{k=1}^{K_L}$ can be used for various tasks, such as predicting the next observation and the properties of underlying processes (*e.g.*, position, velocity).

# 5   Object-Centric Learning with SlotSSM

In this section, we present a concrete example of adapting the generic sequence modeling architecture proposed in Section 4 to solve a specific task. We consider the task of object-centric representation learning from unannotated videos of interacting objects, a typical example of sequences with modular underlying structures. The goal is to obtain a representation for each individual object that captures relevant attributes such as object position, size, shape, color, *etc.* without any object-level annotation.

## 5.1   Object-Centric SlotSSMs (OC-SlotSSMs)

Inspired by previous works [46, 67], we make slight modifications to our sequence modeling architecture to facilitate the discovery of modular structures. We call the resulting model OC-SlotSSMs. First, we use the same number of slots across all layers. It is thus unnecessary to have a slot encoder per layer. However, we find it helpful to still have it, but in another form that encourages iterative refinement of the slots. Specifically, we use the slots output from the previous layer $\{x_{t|l-1}^k\}_{k=1}^K$ as queries, and provide the input tokens $\{x_{t|0}^m\}_{m=1}^M$ as keys and values. Second, we introduce competition among slots in the attention layers of the slot encoder. We achieve this by using inverted attention [61, 67], which is essentially cross attention with the Softmax operation performed over the queries instead of the keys. This has the effect of softly assigning each input token to a slot, thereby promoting modularity. The computation at each layer $l = 1, \ldots, L$ can be summarized as follows:

$$\{\boldsymbol{s}_{t|l}^k\}_{k=1}^K = \text{InvAttn}\Big(\mathbf{q} = \{\boldsymbol{x}_{t|l-1}^k\}_{k=1}^K,\ \mathbf{kv} = \{\boldsymbol{x}_{t|0}^m\}_{m=1}^M\Big)\ , \tag{14}$$

$$\{\boldsymbol{y}_{t|l}^k\}_{k=1}^K,\ \{\boldsymbol{h}_{t|l}^k\}_{k=1}^K = \text{SlotSSM}\Big(\{\boldsymbol{s}_{t|l}^k\}_{k=1}^K,\ \{\boldsymbol{h}_{t-1|l}^k\}_{k=1}^K\Big)\ , \tag{15}$$

$$\{\boldsymbol{x}_{t|l}^k\}_{k=1}^K = \text{SlotMixer}\Big(\{\boldsymbol{y}_{t|l}^k\}_{k=1}^K\Big)\ . \tag{16}$$

We note that the queries in the first inverted attention layer are the learnable CLS tokens $\{\text{CLS}_{t|0}^k\}_{k=1}^K$.

## 5.2 Training Pipeline

Following previous works in object-centric learning [46, 39, 57], we adopt an auto-encoding training pipeline. Given a sequence of video frames $\{\boldsymbol{o}_t \in \mathbb{R}^{H \times W \times 3}\}_{t=1}^T$, we obtain the input $\boldsymbol{x}_{t|0}$ to our sequence modeling architecture by applying a CNN encoder to each frame $\boldsymbol{o}_t$ and adding a positional embedding for each feature map cell. The output slots $\{\boldsymbol{x}_{t|L}^k\}_{k=1}^K$ are each decoded into an object image $\hat{\boldsymbol{o}}_t^k \in \mathbb{R}^{H \times W \times 3}$ and an alpha mask $\boldsymbol{\alpha}_t^k \in \mathbb{R}^{H \times W \times 1}$ by a spatial broadcast decoder [66]. The final reconstruction $\hat{\boldsymbol{o}}_t \in \mathbb{R}^{H \times W \times 3}$ is given by the alpha-composition of the object images:

$$\hat{\boldsymbol{o}}_t^k, \boldsymbol{\alpha}_t^k = \text{Decoder}(\boldsymbol{x}_{t|L}^k)\ , \quad \hat{\boldsymbol{o}}_t = \sum_{k=1}^K \frac{\exp(\boldsymbol{\alpha}_t^k)}{\sum_{j=1}^K \exp(\boldsymbol{\alpha}_t^j)} \cdot \hat{\boldsymbol{o}}_t^k\ . \tag{17}$$

The training objective is to minimize the reconstruction error $\mathcal{L} = \frac{1}{T}\sum_{t=1}^T \|\hat{\boldsymbol{o}}_t - \boldsymbol{o}_t\|_2^2$.

## 6 Related Work

**State Space Models (SSMs).** Popularized by S4 [27], SSMs have attracted growing interest in language modeling and as a sequence modeling framework in general. The original S4 follows the HiPPO theory [25] to parameterize and initialize the state transition matrices, which is quite mathematically involved. Most recent works have proposed simplified versions that use diagonal transition matrices [29, 26, 58] and pure RNN formulation (*i.e.*, without reliance on ODE discretization) [30, 50, 9]. Several works have proposed hybrid architectures of SSMs and Transformers to incorporate their complementary strengths [75, 48, 17, 32, 24]. In addition to language modeling, SSMs have been applied to various domains, including time-series generation [73], audio generation [19], visual classificiation and generation [49, 40, 32, 65, 74, 71], and reinforcement learning [8, 47, 10, 52]. Our study introduces the first SSM with inductive biases for modeling inherently modular processes.

**Object-Centric Learning.** Object-centric learning seeks to discover modular structures and independent mechanisms [20] such as objects and their relations from multi-object images and videos with weak or no supervision [3, 22, 38, 23, 15, 14, 16, 7, 45, 35, 41, 37, 6, 44, 11, 1, 64, 68, 53, 34]. Recent works are predominantly based on the Slot Attention [46] model, which uses a GRU [5] and competitive attention mechanisms to iteratively refine slot representations [54, 57, 39, 13, 69, 55, 36, 70]. However, GRUs and RNNs in general are prone to vanishing gradient issues [51], and the training must be done in a sequential way. These weaknesses render them incapable of scaling up to long-range videos. Additionally, hardware parallelization for video object-centric learning has been explored in [56], however, it incurs quadratic cost unlike ours. Our SlotSSMs framework can be specialized to address object-centric learning tasks effectively. By integrating SSMs at its core, SlotSSMs benefit from parallelizable training and possess remarkable long-term memory capabilities. Moreover, as a versatile framework, SlotSSMs are well-suited to tackle other tasks such as long-range visual reasoning.

## 7 Experiments

We present an extensive evaluation of our models across a variety of tasks. Section 7.1 illustrates the need for modular latent states through a multi-object video prediction task. Section 7.2 demonstrates the advantages of SlotSSMs over Transformers and RNNs using a newly proposed long-context reasoning benchmark. Section 7.3 investigates the object-centric learning capabilities of OC-SlotSSMs. Finally, Section 7.4 showcases the 3D visual reasoning capabilities using the CATER benchmark [18].

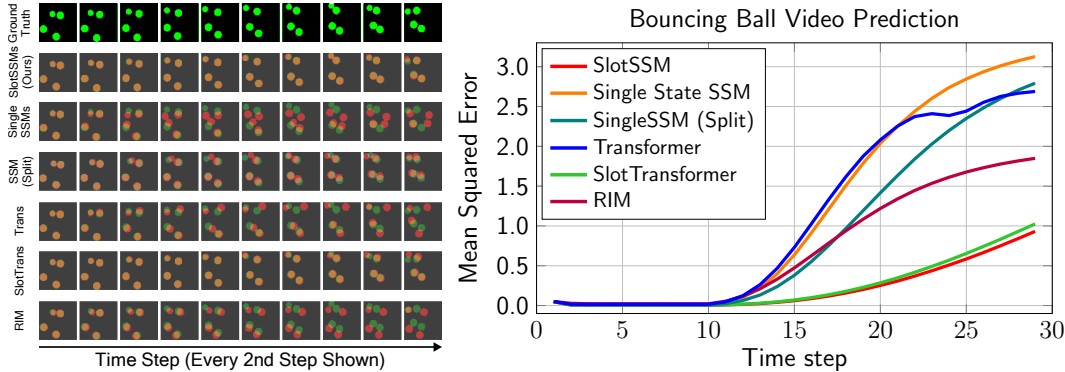

**Figure 4: Multi-Object Video Prediction Task**. *Left*: Generated video frames at every second step, showing 10 of the 20 total frames generated. Green color indicates ground-truth and red color indicates predictions. *Right*: MSE over a 20-frame autoregressive rollout, given 10 context frames. SlotSSM demonstrates its efficiency in modeling multi-object dynamics.

## 7.1 Multi-Object Video Prediction

We begin with a multi-object video modeling task to demonstrate the benefit of incorporating modularity into state space.

**Dataset and Task.** We utilize the bouncing balls video dataset introduced by [62], which consists of videos of white balls bouncing off each other in an empty window. Each ball has random initial positions, velocities, and masses, governing their interactions. The task is conditional video generation, specifically $p(\mathbf{x}_{T+1:T+W}|\mathbf{x}_{1:T})$. This task is inherently modular as it requires models to remember each object's attributes and interaction rules.

**Experimental Setup.** We train models on 20-frame sequences using teacher-forcing and binary cross-entropy loss. At test time, given $T = 10$ context frames, the model autoregressively predicts $W = 20$ future frames using its own outputs. Performance is evaluated using Mean Squared Error (MSE) between predicted and ground-truth images.

**Models.** We employ the SlotSSM architecture described in Section 4.3 We use the same number of slots across layers and apply the Slot Encoder only at the first layer. We compare our model against several baselines: Single State SSM, which shares the same architecture but uses a monolithic state; Single State SSM (Split), which uses a Single State SSM with multi-slot encoder and decoder—slots are concatenated in SSM, then split into multiple slots for the decoder; RIM[20], a slot-based RNN model with separate RNN weights per slot that introduces sparse slot updates and interactions based on input attention values; Transformer, a vanilla Transformer model with a single input embedding per time step; and SlotTransformer, a Transformer model with multiple input slots at each time step.

All models share the same encoder and decoder architectures. The encoder is a Transformer described in Section 4.1, using a single CLS token for single-state models. The decoder consists of three Transformer layers with self-attention for image patches and cross-attention to query the slots. We use six slots for all slot-based models. For RIM, we set $k = 4$ for top-$k$ active modules as in the original paper. We carefully match hyperparameters across baselines to ensure comparable model sizes, except for RIM, which inherently requires a larger model due to separate RNN weights per slot. Additional implementation details are in Appendix C.

**Results.** Figure 4 compares model performances, showing that SlotSSM outperforms all baselines, including a slight improvement over SlotTransformer. The significant gap between SlotSSM and Single State SSM underscores the importance of modular slot states for effective multi-object dynamics learning, as also evidenced by the comparison between Transformer and SlotTransformer. SlotSSM also significantly outperforms Single State SSM (Split), which uses the same modular encoder and decoder, highlighting that modularity in temporal modeling—the core contribution of SlotSSM—is critical for improved performance. While the RIM model performs better than other single-state baselines, it still lags behind SlotSSM and SlotTransformer.

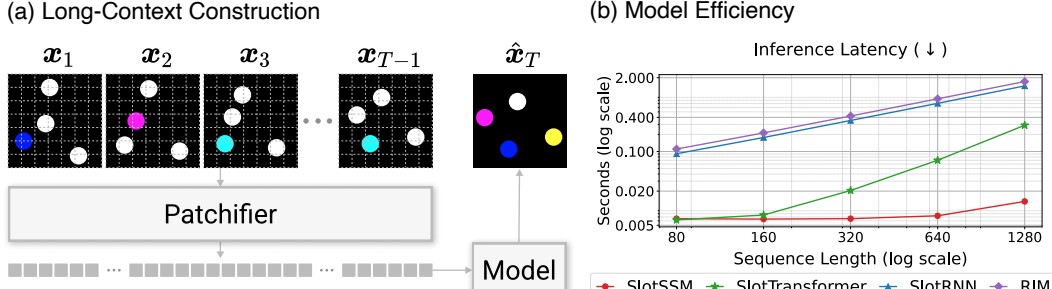

**Figure 5: Long-Context Construction and Model Efficiency in the Blinking Color Balls Benchmark.** *Left*: We construct long-sequence inputs by patchifying the context images. *Right*: Comparison of model inference latency with batch size 6. SlotSSM demonstrates computational efficiency for long-sequence processing tasks.

## 7.2 Long-Context Reasoning

We now evaluate the long-context reasoning capabilities of SlotSSM. To enable a rigorous assessment in a multi-object setting, we propose the novel Blinking Color Balls Benchmark.

**Blinking Color Balls Benchmark.** This benchmark has two variants—Earliest Color and Most Frequent Color—each consisting of image sequences with context images $\mathbf{x}_{1:T-1}$ and a target image $\mathbf{x}_T$. In each context image, one ball is randomly selected and assigned a non-white color from five options while others remain white. The coloring of balls in the target image $\mathbf{x}_T$ depends on the variant: in **Earliest Color**, each ball's color is its earliest assigned non-white color in the context (remaining white if none); in **Most Frequent Color**, each ball's color is the non-white color assigned most frequently during the context (ties broken by earliest assignment; remaining white if none).

To create a long-range reasoning task, we further patchify each context image into non-overlapping $P \times P$ patches and flatten them into a sequence of length $P^2$, as shown in Figure 5(a). With context length $T - 1$, the total input sequence length is $L = (T - 1) \times P^2$. Models must identify and track objects from partial patch views while remembering and counting color assignments, making the task highly challenging. The final task is to predict the target image given this long sequential input.

**Experimental Setup.** We evaluate models on Earliest Color with $T = 6$ and Most Frequent Color with $T \in \{6, 11\}$, using patch sizes $P \in \{4, 8, 16\}$. This yields input sequence lengths $L \in \{80, 160, 320, 640, 1280, 2560\}$. The Most Frequent Color with $T = 11$ requires stronger memorization and reasoning capabilities due to longer context and more color assignments, .

**Models.** We employ the same encoder, decoder, and the SlotSSM architectures as in Section 7.1. For slot encoding, each image patch is treated as an image and processed by the slot encoder. The slots from the last time step are provided to the decoder to predict the full target image. We compare our SlotSSM against several baselines: Single State SSM, SlotTransformer, and RIM. Additionally, we introduce a novel slot-based design called SlotRNNs, which shares RNN weights across slots and uses self-attention layers between time steps as the slot mixer. SlotRNNs can be viewed as a special case of RIMs with shared weights and dense state updates. Empirically, SlotRNNs exhibit more stable training and improved performance compared to RIMs. For fair comparison, all slot-based models use six slots, and we carefully match model sizes as in Section 7.1.

**Results.** Figure 6 shows that SlotSSM outperforms Single State SSM, SlotRNN, and RIM across all sequence lengths. For shorter sequences (80 and 160), Single State SSM and SlotRNN have relatively low error rates but degrade significantly beyond 320 frames. Surprisingly, RIM fails to generalize at any sequence length, likely due to optimization issues from separate weights per slot; our SlotRNN addresses this by sharing weights across slots while maintaining modularity. SlotTransformer performs competitively up to 640 frames. However, SlotSSM demonstrates superior long-range reasoning, especially at 1280 and 2560 frames, where other models cannot run due to memory or optimization constraints. Figure 5(b) highlights SlotSSM's computational efficiency. SlotTransformer's inference latency increases rapidly with sequence length due to quadratic complexity, SlotSSM maintains stable and efficient inference across all lengths. Due to SlotTransformer's high memory usage, we used a batch size of 6 for latency evaluation. Qualitative comparisons in Appendix B.3 provide further insights into the models' strengths and weaknesses.

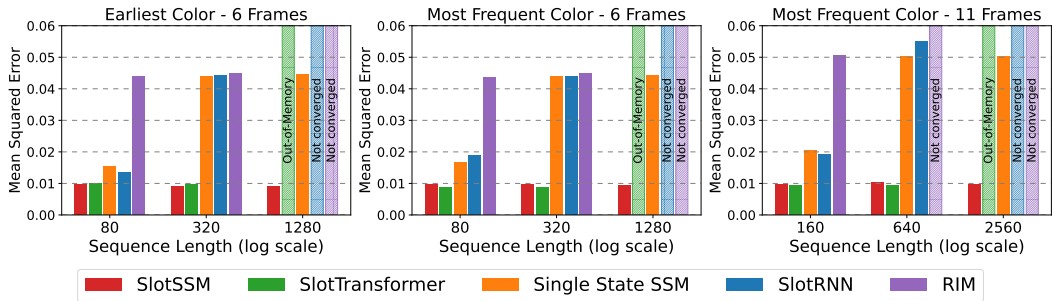

**Figure 6: Long-Context Reasoning in Blinking Balls Benchmark.** SlotSSM maintains consistent performance across sequence lengths from 80 to 2560, whereas baseline models show degraded performance or fail to complete training due to high memory and computational requirements.

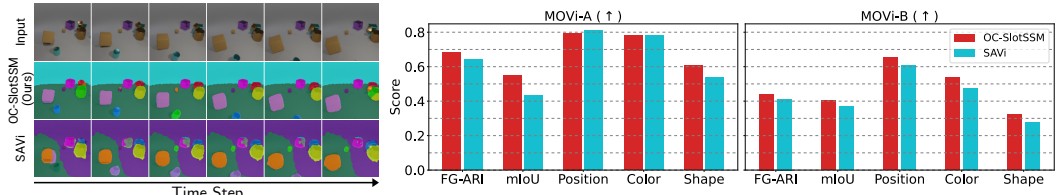

**Figure 7: Object-Centric Learning Results.** *Left*: Qualitative comparison of segmentation masks on MOVi-A. OC-SlotSSM demonstrate less object spliting and better boundary adherence. *Right*: Quantitative evaluation on unsupervised object segmentation and attribute prediction. OC-SlotSSM outperforms SAVi on most metrics.

## 7.3  Unsupervised Object-Centric Learning

In this section, we evaluate the performance of the Object-Centric SlotSSMs (OC-SlotSSM) variant in unsupervised object-centric representation learning.

**Datasets.** We evaluate OC-SlotSSM on the MOVi-A and MOVi-B subsets of the MOVi video dataset [21], which contain videos of up to 10 objects moving in a 3D environment. MOVi-B adds complexity over MOVi-A by including a wider variety of object types and multi-colored backgrounds.

**Tasks.** Following prior object-centric learning works [46, 36], we evaluate models on two downstream tasks: unsupervised object segmentation and attribute prediction. For segmentation, we report FG-ARI and mIoU metrics. For attribute prediction, we measure the quality of representations by inferring object properties: we report prediction accuracy for discrete attributes (e.g., object shape) and $R^2$ for continuous attributes (e.g., object position).

**Models.** We compare OC-SlotSSM to SAVi [39], an RNN-based object-centric learning approach. Both models use a CNN encoder to extract image features as input tokens $x_t \mid 0^m m = 1^M$, which are processed by their respective attention mechanisms—inverted attention in OC-SlotSSM and slot attention in SAVi—to produce slots. These slots are then used to reconstruct the image and generate per-object segmentation masks via a spatial broadcast decoder, with reconstruction as the training objective. For unsupervised object segmentation, we directly use the object masks obtained during training. For attribute prediction, we match slots to object IDs using Hungarian matching based on segmentation masks, then use linear heads and 2-layer MLPs to predict discrete and continuous attributes, respectively, keeping the slots frozen.

**Results.** Results in Figure 7 demonstrate that OC-SlotSSM consistently outperforms SAVi in unsupervised object segmentation on both MOVi-A and MOVi-B. The qualitative comparison (Figure 7, left) shows that OC-SlotSSM generates masks with tighter object boundaries and fewer object splitting, which also leads to improved attribute prediction accuracy (Figure 7, right). Furthermore, we empirically found that OC-SlotSSM exhibits superior stability during training compared to SAVi, which tends to collapse into a single slot representing the entire scene when trained long enough. This collapse is not reflected in the validation loss, so we apply early stopping based on manual inspection. In contrast, OC-SlotSSM does not suffer from this instability, demonstrating its robustness in learning object-centric representations.

Table 1: Performance on CATER Snitch Localization Task.

| Model | No Pre-train | | Pre-train | |
|---|---|---|---|---|
| | Top-1 Acc (%) | Top-5 Acc (%) | Top-1 Acc (%) | Top-5 Acc (%) |
| Single State SSM | 10.27 | 27.21 | 41.15 | 65.70 |
| SlotTransformer | 41.09 | 62.24 | 49.21 | 70.24 |
| SlotSSM | 25.64 | 45.03 | 54.73 | 74.42 |
| OC-SlotSSM | **61.58** | **84.00** | **69.27** | **90.48** |

## 7.4 3D Visual Reasoning

Finally, we explore the application of SlotSSM and OC-SlotSSM to 3D visual reasoning tasks using the CATER benchmark [18].

**CATER Benchmark.** CATER consists of 300-frame video episodes of objects moving in a 3D environment. The movement can lead to partial occlusions and even complete coverage of smaller objects by larger ones. The primary task is snitch localization—predicting the golden snitch's location in the final frame. The snitch is always present but may be occluded. Models must reason about its location based on the last visible position and other objects' movements. Success in this task demonstrates models' capacity for complex visual reasoning in dynamic 3D environments.

**Experimental Setup.** We consider two experiment settings: direct training and pre-training + fine-tuning. In direct training, models are trained end-to-end on the snitch localization task. In pre-training + fine-tuning, models are first pre-trained on video inputs using a reconstruction objective, then fine-tuned on the task-specific signal. During pre-training, we randomly sample 32 frames from the 300-frame videos. For direct training and fine-tuning, we split the sequence into 50 non-overlapping segments of 6 frames each, randomly selecting one frame from each to create a 50-frame sequence spanning the entire video. At test time, we evenly sample 50 frames by skipping every 6 frames. The snitch's final location is quantized into a 6×6 grid, framing the problem as a classification task.

**Models.** We evaluate the performance of SlotSSM, OC-SlotSSM, Single State SSM, and Slot-Transformer. We exclude RNN-based baselines, as our preliminary experiments reveal that they are unstable when handling long video inputs and prone to collapse to a constant output. For the visual pre-training setting, we employ a spatial broadcast decoder to reconstruct the input images. During downstream training/fine-tuning, we feed the slots from the final step to a transformer predictor with single CLS token, followed by a linear layer on the output CLS token to predict the snitch's position.

**Results.** Table 1 presents the Top-1 and Top-5 accuracy on the CATER Snitch Localization task. Consistent with our previous findings, SlotSSM outperforms Single State SSM, highlighting the importance of modular latent structures. Comparing SlotSSM with SlotTransformer, we see notable differences between direct training and pre-training settings: in direct training, SlotTransformer surpasses SlotSSM, possibly due to optimization advantages from direct access to all previous states; however, SlotSSM benefits more from pre-training, likely due to the explicit memory capacity of SSM states, consequently, pre-trained SlotSSMs outperforming their SlotTransformer counterparts.

Remarkably, OC-SlotSSM achieves the highest accuracy, outperforming all baselines by a large margin in both direct training and pre-training settings. This performance gain may be attributed to the explicit decomposition into object-centric representations, which facilitates reasoning about object properties, relationships, and interactions.

## 8 Conclusion

In this work, we presented SlotSSMs a novel approach to incorporating modular structure and inductive biases into State Space Models for improved sequence modeling. By maintaining a collection of independent slot vectors and performing state transitions independently per slot with sparse interactions via self-attention, SlotSSMs effectively captures the inherent modularity present in many real-world processes. The experimental results in object-centric video understanding and video prediction tasks demonstrate the substantial performance gains offered by SlotSSMs over existing sequence modeling methods.

## Acknowledgements

This research was supported by GRDC (Global Research Development Center) Cooperative Hub Program (RS-2024-00436165) and Brain Pool Plus Program (No. 2021H1D3A2A03103645) through the National Research Foundation of Korea (NRF) funded by the Ministry of Science and ICT.

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

# A    Limitations & Broader Impact

**Limitations** SlotSSMs' success illustrates the importance of designing architectures that align with the problem domain's underlying modular structure. It also paves the way for future research in modular and object-centric sequence modeling. However, it has some limitations that future studies could address. First, compared to Transformer architectures, we find that SlotSSMs could benefit more from a pre-training phase in visual reasoning tasks. For example, in the 3D visual reasoning task, SlotSSMs underperform Transformer models when trained without pretraining. However, when combined with task-free pretraining, SlotSSMs demonstrate significant improvement, enabling them to outperform Transformer models. We note that this effect of task-free pre-training is more prominent in SlotSSMs than in Transformer baselines. This suggests that for tasks with sparse training signals, the sequential nature of SlotSSM performs better with a pre-training phase to learn to effectively utilize information from all time steps. We believe this phenomenon is worth further investigation in future research. Second, while the proposed architecture is applicable beyond video modeling—for example, to other modalities such as text and audio—this study has not explored these possibilities. It remains a matter for future work. Thrid, due to our academic research lab's computing resource constraints, we were unable to significantly scale up the proposed model to industry-scale in terms of model size and data size. Scaling up SlotSSMs could uncover additional properties or limitations that are not evident at the current scale of experimentation. Lastly, future studies should investigate the effect of increased visual complexity in videos. As an initial step, we present a preliminary exploration in Appendix D.2, where SlotSSMs are applied to natural video scenes. These experiments illustrate how modularity can emerge through the independent mechanisms of SlotSSMs in real-world scenarios. We hope these findings will inspire future research on the industry-scale adoption of SlotSSMs.

**Impact Statement** The introduction of SlotSSMs, a novel framework that incorporates independent mechanisms into State Space Models (SSMs), has the potential to significantly impact the field of sequence modeling. By leveraging the modular structure inherent in many real-world processes, SlotSSMs offers a more intuitive and effective approach to modeling long-range temporal dependencies in object-centric video understanding and prediction tasks. The substantial performance gains demonstrated by SlotSSMs over existing sequence modeling methods highlight the importance of designing architectures that align with the underlying structure of the problem domain. This breakthrough could lead to the development of more efficient and accurate models for a wide range of applications, such as robotics, autonomous vehicles, and video surveillance systems. Moreover, the success of SlotSSMs in capturing the modular nature of real-world processes could inspire further research into modular and object-centric sequence modeling. This could result in the development of even more advanced architectures that can better handle the complexity and diversity of real-world data. Because this is a general backbone architecture for sequence modeling, it doesn't raise direct ethical concerns. However, its ethical implications depend on the way downstream application developers use the model.

# B    Blinking Color Balls Benchmark

## B.1    Motivation

Real-world videos are often inherently modular, involving multiple dynamic entities and their interactions across time. However, existing long-range reasoning tasks, such as those in the Long-Range Arena Benchmark [60], are typically designed to focus on single-object settings and recognizing a single dynamic pattern in the observations. To bridge this gap and facilitate more comprehensive evaluation, we propose the Blinking Color Balls Benchmark, a long-range visual reason benchmark desgined in a multi-object setting.

## B.2    Dataset Design

We provide an illustrative example of the dataset design in Figure 8. Each episode of the dataset contains a context-target pair $(\mathbf{x}_{1:T-1}, \mathbf{x}_T)$. At each timestep in $\mathbf{x}_{1:T-1}$, all bouncing balls are first colored white, and then one ball is randomly picked and colored with one of 5 non-white colors. This process is repeated for all context frames, and it is represented in the rows in Figure 8(top). Note that the object picking and coloring are performed independently for each timestep, thus one ball could

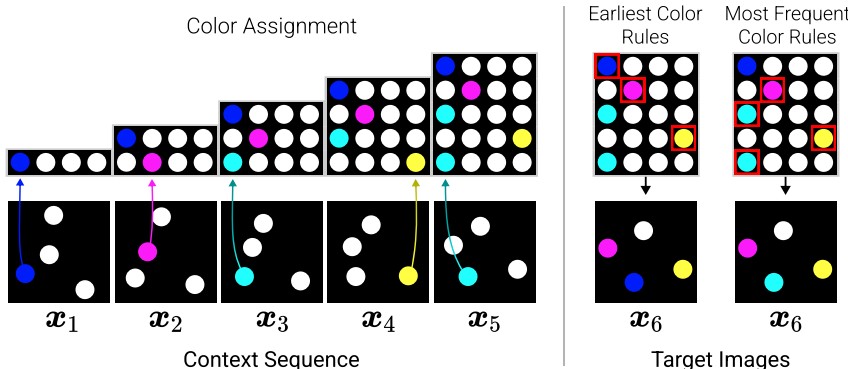

**Figure 8: Blinking Color Balls Benchmark Overview.** *Left*: Context frames with independent random ball picking and color assignments for each frame. Top figures indicate the sequential color assignment. *Right*: Target image for the Earliest Color and Most Frequent Color variants. Top figures indicate the color assignment rules.

be selected none or multiple times and colorized with the same or different colors across different timesteps.

The target images are then constructed with two rules: Earliest Color and Most Frequent Color. The Earliest Color rule picks the earliest non-white color assigned to the ball as the final color, while the Most Frequent Color rule counts the assignment of each non-white color and picks the color with the highest count (if there are ties, the earlier color among the highest is chosen). In Figure 8, we differentiate the two datasets using the same context sequence, which will result in different target images based on the rule. Note that regardless of the color assignment, the objects are moving and follow the physical bouncing rules throughout the full sequence. More image samples can be found in Figure 9.

Finally, as illustrated in Figure 5(a), we transform the conditional image generation task into a long-range reasoning task by using patchified context images as input. Instead of providing the $T - 1$ context images directly to the model, we flatten non-overlapping patches of the original images to create a long input sequence. Given $P \times P$ patches per image, the context length becomes $L = (T - 1) \times P^2$. Note that patchification is used intentionally to construct long sequences for the benchmark; SlotSSMs in general do not inherently require patchified inputs and instead use a Slot Encoder to extract slots as input at each time step.

### B.3 Challenges and Qualitative Comparison

The Blinking Color Balls tasks pose significant challenges for the models, as they are required to reason about the object movement and color assignment rules from partial views of objects in temporally distant image patches. We can define two levels of challenges: (1) identifying the objects from image patches and predicting their future positions based on their dynamics, and (2) determining the final color assignment of each object based on the given rules. The first challenge is relatively simple, as it primarily involves learning the dynamics of objects from the past two frames prior to the target time step. However, the second challenge is particularly difficult, as it requires the model

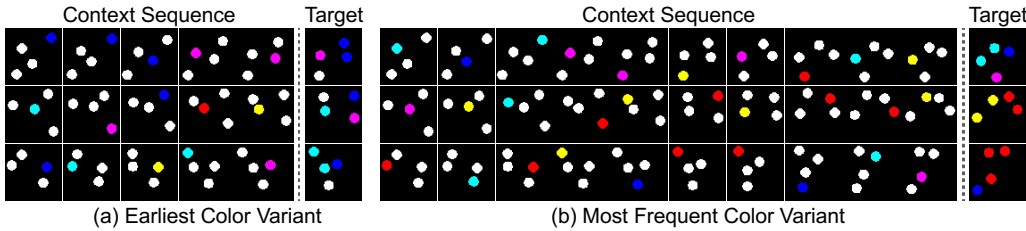

**Figure 9: Blinking Color Balls Samples.**

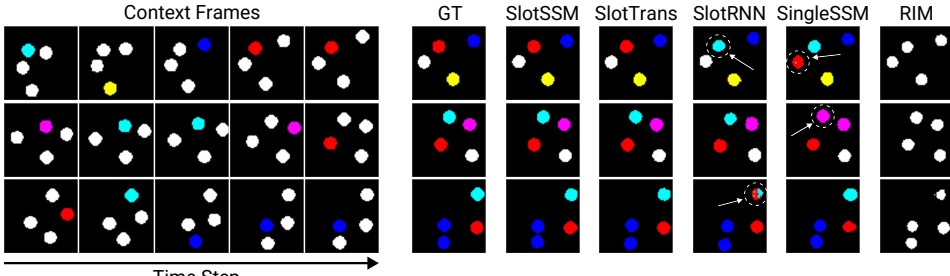

**Figure 10: Blinking Color Balls Qualitative Comparison.** Results shown for the Most Frequent Color variant with a sequence length of 80 frames.

to reason over the entire input sequence, necessitating the identification of an object's history from partially observed patches in a long-range context.

Figure 10 presents a qualitative comparison of the models' performance on the task. The results reveal a clear categorization of the models based on their capability to address the two levels of challenges. The baseline RIM model successfully predicts the object positions in the target image but struggles with learning the color assignment rules. Consequently, it predicts the color white that generally have the highest appearance probability for all objects. Note that the rendered images are based on the argmax of the logits over the color categories. Models such as SlotRNN and Single State SSM demonstrate the ability to learn color assignments, but they make mistakes in some cases. In contrast, SlotSSM and SlotTransformer successfully achieve both accurate position prediction and color assignment.

## C  Additional Implementation Details

### C.1  SlotSSMs and OC-SlotSSMs

**Slot Encoder.** The main difference between the SlotSSMs and OC-SlotSSMs variants is in the design of the *Slot Encoders* as illustrated in Figure 11. The Slot Encoder in SlotSSMs is implemented as a multi-layer transformer with self-attention and cross-attention modules. Given the input tokens $\mathcal{X}_t = \{x_t^m\}_{m=1}^M$, the structure of each layer in the Slot Encoder can be delineated into three modules:

$$\mathcal{C}_t = \text{SelfAttn}(\mathcal{C}_t) \, , \tag{18}$$

$$\mathcal{C}_t = \text{CrossAttn}\left(\texttt{q} = \mathcal{C}_t, \ \texttt{kv} = \mathcal{X}_t\right) \, , \tag{19}$$

$$\mathcal{C}_t = \text{MLP}(\mathcal{C}_t) \, . \tag{20}$$

We use 3 layers in all our experiments. Note that we also apply skip connections and layer normalization in the input for all three modules, but have omitted them in the equations for brevity. The regular cross-attention used here employs softmax normalization over the attention weights applied to the input tokens:

$$Q = W_Q(\mathcal{C}_t), \quad K = W_K(\mathcal{X}_t), \quad V = W_V(\mathcal{X}_t) \, , \tag{21}$$

$$\mathcal{C}_t^{\text{out}} = \texttt{softmax}\left(\frac{QK^T}{\sqrt{D}}, \quad \texttt{axis='keys'}\right) V \, . \tag{22}$$

In the OC-SlotSSMs layers, the *Slot Encoder* is implemented as a single inverted attention layer. This layer differs from the regular cross attention by the way attention weights are normalized:

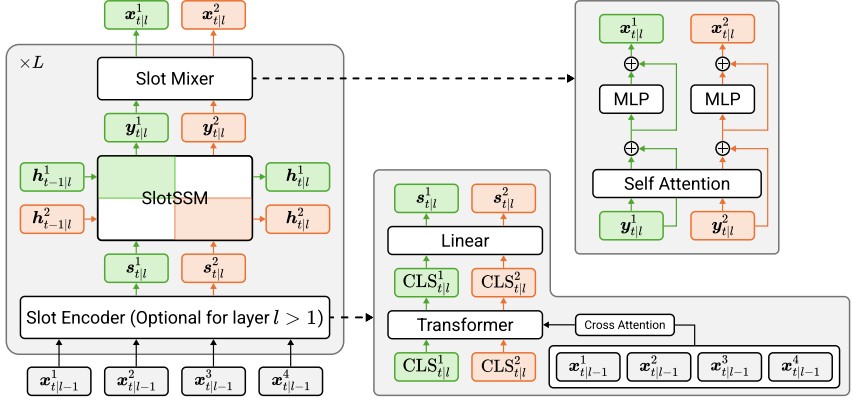

(a) SlotSSMs

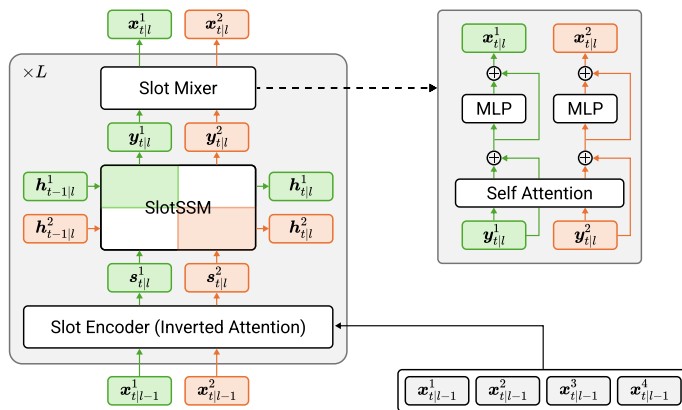

(b) OC-SlotSSMs

**Figure 11: SlotSSMs vs OC-SlotSSMs.**

$$Q = W_Q(\mathcal{C}_t), \quad K = W_K(\mathcal{X}_t), \quad V = W_V(\mathcal{X}_t)\,, \tag{23}$$

$$A = \texttt{softmax}\left(\frac{QK^T}{\sqrt{D}}, \quad \texttt{axis=`queries'}\right)\,, \tag{24}$$

$$A_{i,j} = \frac{A_{i,j}}{\sum_{j=1}^{N_K} A_{i,j}}\,, \tag{25}$$

$$\mathcal{C}_t^{\text{out}} = AV\,. \tag{26}$$

The inverted attention layer applies softmax normalization over the queries, introducing a competition among the query tokens over the attention to the input tokens and thereby promoting disentanglement for the input tokens.

**SSM Blocks.** For the implementation of the SSM models, we leverage recent advances in linear state space models and design our SSM block in SlotSSM based on the Mamba architecture [24]. The block-diagonal transition of slots is implemented as parallel runs of SSM blocks that share the same model weights.

$$\{\boldsymbol{y}_{t|l}^k\}_{k=1}^{K_l},\ \{\boldsymbol{h}_{t|l}^k\}_{k=1}^{K_l} = \text{SlotSSM}\Big(\{\boldsymbol{s}_{t|l}^k\}_{k=1}^{K_l},\ \{\boldsymbol{h}_{t-1|l}^k\}_{k=1}^{K_l}\Big) \tag{27}$$

$$\implies \quad \boldsymbol{y}_{t|l}^k, \boldsymbol{h}_{t|l}^k = \text{MambaBlock}\Big(\boldsymbol{s}_{t|l}^k, \boldsymbol{h}_{t-1|l}^k\Big), \quad \forall k \in \{1, \ldots, K_l\} \tag{28}$$

| Module | Hyperparameter | Dataset & Models | |
| --- | --- | --- | --- |
| | | Blinking Color Balls (SlotSSMs) | MOVi-A (OC-SlotSSMs) |
| General | Batch Size | 128 | 24 |
| | Training Steps | 300K | 500K |
| | Sequence Length | {80, 160, 320, 640, 1024, 2048} | 6 |
| | Optimizer | AdamW | AdamW |
| | Weight Decay | 0.1 | 0.1 |
| | Learning Rate | 8e-4 | 3e-4 |
| Slot Encoder | Input Tokenizer | $\mathrm{MLP}(\mathrm{Patchify}(\boldsymbol{x}_{\mathrm{input}}))$ | $\mathrm{Flatten}(\mathrm{CNN}(\boldsymbol{x}_{\mathrm{input}}))$ |
| | Encoder Type | Self-Cross Attention | Inverted Attention |
| | Applied Layers | First Layer | All Layers |
| | Hidden Size | 64 | 192 |
| | Dropout | 0 | 0 |
| | Heads | 4 | 4 |
| SlotSSM | Hidden Size | 64 | 192 |
| | # Slots | 6 | 11 |
| | SSM Model | Mamba Block | Mamba Block |
| | State Size | 16 | 16 |
| | State Expand | 1.25 | 1.25 |
| Slot Mixer | Dropout | 0 | 0 |
| | Heads | 4 | 4 |

**Table 2:** Hyperparameters of our model used in our experiments.

We include pseudo-code of the Mamba block implementation in Algorithm 1. For a more detailed description of the Mamba architecture and its underlying principles, we refer the readers to the original paper [24].

## C.2 Baseline Models

We use the official implementation of RIM from GitHub [3], as well as the SAVi implementation from STEVE [4]. We describe the implementation of the proposed baselines SlotRNN and SlotTransformer in the following.

**SlotRNN.** SlotRNN adopts a similar design to SlotSSM, but replaces the SSMs with GRUs [5]. In this architecture, the slots are processed in parallel across different slots at each time step and sequentially across time steps. The implementation of each layer is summarized as follows.

$$\{\boldsymbol{s}_{t|l}^k\}_{k=1}^{K_l} = \mathrm{SlotEncoder}\Big(\{\boldsymbol{x}_{t|l-1}^k\}_{k=1}^{K_{l-1}}\Big) \ , \tag{29}$$

$$\boldsymbol{h}_{t|l}^k = \mathrm{GRU}\Big(\boldsymbol{s}_{t|l}^k, \boldsymbol{h}_{t-1|l}^k\Big), \quad \forall k \in \{1, \ldots, K_l\} \ , \tag{30}$$

$$\{\boldsymbol{h}_{t|l}^k\}_{k=1}^{K_l} = \mathrm{SelfAttention}\Big(\{\boldsymbol{h}_{t|l}^k\}_{k=1}^{K_l}\Big) \ , \tag{31}$$

$$\{\boldsymbol{x}_{t|l}^k\}_{k=1}^{K_l} = \{\boldsymbol{h}_{t|l}^k\}_{k=1}^{K_l} \tag{32}$$

**SlotTransformer.** SlotTransformer uses the same SlotEncoder as SlotSSM to obtain slot representations. At each time step, the slots from the current step are concatenated with the slots from all previous time steps. This combined sequence is then processed using a Transformer with causal mask in time dimension which ensures that each slot can only obtain information from prior or current time steps. The implementation of each layer is summarized as follows:

---
[3]https://github.com/anirudh9119/RIMs
[4]https://github.com/singhgautam/steve

**Table 3:** The CNN encoder architecture used for object-centric learning.

| Layer | Kernel Size | Stride | Padding | Channels | Activation |
|-------|-------------|--------|---------|----------|------------|
| Conv | $5 \times 5$ | 2 | 2 | 192 | ReLU |
| Conv | $5 \times 5$ | 1 | 2 | 192 | ReLU |
| Conv | $5 \times 5$ | 1 | 2 | 192 | ReLU |
| Conv | $5 \times 5$ | 1 | 2 | 192 | None |

**Table 4:** Spatial broadcast decoder architecture for image reconstruction in object-centric learning, it outputs RGB and alpha-mixing logits.

| Layer | Kernel Size | Stride | Padding | Channels | Activation |
|-------|-------------|--------|---------|----------|------------|
| Slot Normalization | - | - | - | - | - |
| Positional Embedding | - | - | - | - | - |
| ConvTranspose2d | $5 \times 5$ | 2 | 2 (Output Padding: 1) | 64 | ReLU |
| ConvTranspose2d | $5 \times 5$ | 2 | 2 (Output Padding: 1) | 64 | ReLU |
| ConvTranspose2d | $5 \times 5$ | 2 | 2 (Output Padding: 1) | 64 | ReLU |
| ConvTranspose2d | $5 \times 5$ | 2 | 2 (Output Padding: 1) | 3 + 1 | None |

$$\{s_{t|l}^k\}_{k=1}^{K_l} = \text{SlotEncoder}\left(\{x_{t|l-1}^k\}_{k=1}^{K_{l-1}}\right) , \tag{33}$$

$$\{x_{<=t|l}^k\}_{k=1}^{K_l} = \text{Transformer}\left(\{s_{t|l}^k\}_{k=1}^{K_l} \cup \{s_{<t|l}^k\}_{k=1}^{K_l}\right) , \tag{34}$$

### C.3 Blinking Color Balls Experiemtns

We show the hyperparameters used in the experiments in Table 2.

**Input Tokenizer.** Each patch in the input sequence is treated as an image and further split into non-overlapping patches of size $4 \times 4$. Each patch is then augmented with spatial and temporal positional embeddings, followed by an MLP layer to compute the final tokens for the Slot Encoder.

**Decoder.** During image decoding, we use a self-cross attention layer with positional embeddings as input and slots as context. Given the positional embeddings $\mathcal{P}_t = \{p_t^m\}_{m=1}^{HW}$ and slots from SlotSSM $\mathcal{S}_t = \{s_t^k\}_{k=1}^K$, each layer of the transformer decoder can be described as follows:

$$\mathcal{P}_t = \text{SelfAttn}(\mathcal{P}_t) , \tag{35}$$

$$\mathcal{P}_t = \text{CrossAttn}\left(\texttt{q} = \mathcal{P}_t, \texttt{kv} = \mathcal{S}_t\right) \tag{36}$$

$$\mathcal{P}_t = \text{MLP}(\mathcal{P}_t) . \tag{37}$$

We use a total of 3 layers, and the final pixel logits are computed using a linear head.

**Training Objective.** During training, we transform the image prediction problem into a pixel-wise classification task. Specifically, for a target image $x_N \in \mathbb{R}^{H \times W \times 3}$, we compute a quantization by categorizing each pixel into one of 7 discrete color categories:

$$x_N^Q(i,j) = Q(x_N(i,j)) \quad \forall \, i \in \{1, 2, \ldots, H\}, \, j \in \{1, 2, \ldots, W\} \tag{38}$$

where $Q : \mathbb{R}^3 \to \mathbb{C}$ is the quantization function that maps a 3-dimensional color vector to one of the 7 color categories in the set $\mathbb{C} = \{c_1, \ldots, c_7\}$. Each $c_k \in \mathbb{R}^3$ represents a color vector corresponding to a discrete color category. This is a lossless quantization process since the raw images are generated with the same set of discrete colors. The final training objective is the cross-entropy loss between the model output $\hat{x}_N$ and the target $x_N^Q$:

$$\mathcal{L} = -\sum_{i=1}^{H} \sum_{j=1}^{W} \sum_{k=1}^{6} \boldsymbol{x}_N^Q(i,j,k) \log(\hat{\boldsymbol{x}}_N(i,j,k)) \tag{39}$$

### C.4 Unsupervised Object-Centric Learning Experiments

The hyperparameters used in the experiments are presented in Table 2. Table 4 details the structure of the spatial broadcast decoder described in Section 5.2.

To compute the input tokens, the input images are first processed by a CNN network to generate a 2D feature map. The architecture of the CNN network is described in Table 3. We use a downsampling factor of 2, resulting in an output 2D feature map of size $64 \times 64$ for an input image size of $128 \times 128$. The 2D feature map is then flattened into a sequence of length 4096 and provided to the inverted attention mechanism.

## D  Additional Results

### D.1 Emerging Modularity in SlotSSMs

To gain further insights into the learned representations of the slot-based models, we investigate how the slots are utilized in the image generation process. This can be done by visualizing the attention mechanisms in the decoders.

Figure 12 presents the results of this analysis. For the transformer decoders used in the video prediction and blinking color balls tasks, we compute the argmax over the slots in the cross-attention map (Eq. 36), which represents the attention of the positional tokens over the slots employed to obtain information for reconstruction at each position. In the case of the spatial broadcast decoder, we take the argmax over the alpha-mixing logits $\boldsymbol{\alpha}_t$ (Eq. 17). The visualizations reveal that each slot tends to specialize in representing a specific object or a coherent part of the scene. This emerged object-centric representation allows the model to efficiently capture the dynamics and interactions of the objects, leading to improved performance in tasks such as video prediction and reasoning in the blinking color balls benchmark.

Interestingly, even though the slot encoder used in the video prediction and blinking color balls benchmarks does not explicitly enforce spatial disentanglement constraints like the inverted attention mechanism in OC-SlotSSMs does, the models still learn to represent the sequences in an object-centric manner. This emergent modularity suggests that the SlotSSM design can naturally encourages the model to discover and exploit the underlying structure of the data which is a crucial capability for modeling complex visual inputs such as real-world videos.

### D.2 Real-World Videos Depth Estimation

In this additional experiment, we conduct a preliminary evaluation of SlotSSMs to assess its performance in real-world videos with increased visual complexity. Our aim is to observe how SlotSSMs utilize the modular representations to interpret and process real-world video data. Following previous works in object-centric learning [13], we evaluate this through a depth estimation task.

**Datasets and Tasks.** We select three datasets that represent distinct real-world application scenarios to observe the behavior of SlotSSMs across diverse contexts:

1. **TikTok Dancing Dataset** [33]: Commonly used for content creation tasks [31], this dataset comprises dynamic videos of individuals dancing with variety of movements and poses.

2. **UT Egocentric Video Dataset** [42]: Often utilized for egocentric action recognition [72], this dataset consists of first-person view videos that involve interactions with objects in the environment.

3. **Waymo Open Dataset** [59]: Primarily used for autonomous driving applications [43], this dataset includes videos captured from autonomous vehicles navigating traffic scenarios with diverse environmental conditions.

The primary task across these datasets is to estimate the depth of each pixel in the video frames. However, it is important to emphasize that our objective is not to develop depth estimation models that compete with existing specialized approaches. Instead, our main focus is to use this task, manageable with our lab resources, to showcase the emerging modularity in SlotSSMs for real-world video inputs.

**Models.** We compare OC-SlotSSM, which uses inverted attention in the Slot Encoder, against SAVi++ [13], an RNN-based object-centric learning method. Similar to the setting in Section 5, both models use a CNN encoder to extract input tokens, which are processed using inverted attention for OC-SlotSSM and slot attention for SAVi++ to produce slot representations. These slots are then used to reconstruct the image using a spatial broadcast decoder, with MSE loss as the training objective.

**Results.** The quantitative results in Table 5 show that OC-SlotSSM consistently outperforms the SAVi++ baseline across all datasets, demonstrating its superior video modeling capabilities. More importantly, as illustrated by the attention patterns in Figure 13, unsupervised scene decomposition emerges during training. This demonstrates that SlotSSM is able to utilize the modular representations to discover and exploit the latent structure of the input to complete the task, while the SAVi++ baseline does not demonstrate the same level of emergent modularity.

**Table 5: Depth Estimation MSE ($\downarrow$) on Real-World Datasets.**

| Model | UT Egocentric | Waymo | TikTok |
|---|---|---|---|
| SAVi++ | 0.589 | 0.804 | 1.412 |
| OC-SlotSSM (Ours) | **0.464** | **0.653** | **1.180** |

**Algorithm 1 Mamba Block.** The algorithm receives a $T$-length sequence of the same slot across time $\mathbf{s}_{1:T} \in \mathbb{R}^{T \times D}$. The algorithm outputs the updated slots $\mathbf{s}_{1:T}$. Note that the model imposes the diagonal structure on the $\mathbf{A}$ matrix.

1: **Input**: $\mathbf{s} \in \mathbb{R}^{T \times D}$
2: **Block params**: SSM linear $\mathtt{S}_B, \mathtt{S}_C, \mathtt{S}_\Delta$; Transition matrix $\mathbf{A} \in \mathbb{R}^{D \times N}$; LayerNorm $\mathtt{LN}$; Linear $\mathtt{Linear}_1, \mathtt{Linear}_2$; 1D Conv $\mathtt{Conv1D}$
3:   **for** $t = 1 \ldots T$ **in parallel**
4:     $\mathbf{s}_t, \mathbf{res_t} = \mathtt{Linear}_1(\mathbf{s}_t)$
5:     $\mathbf{s}_t = \mathtt{SiLU}(\mathtt{Conv1D}(\mathbf{s}_t))$
6:   **SSM block**:
7:     $\mathbf{B} \in \mathbb{R}^{T \times N} \leftarrow \mathtt{S}_B(\mathbf{s})$
8:     $\mathbf{C} \in \mathbb{R}^{T \times N} \leftarrow \mathtt{S}_C(\mathbf{s})$
9:     $\Delta \in \mathbb{R}^{T \times D} \leftarrow \mathtt{SoftPlus}(\text{Parameter} + \mathtt{S}_\Delta(\mathbf{s}))$
10:     $\bar{\mathbf{A}}, \bar{\mathbf{B}} \in \mathbb{R}^{T \times D \times N} \leftarrow \text{discretize}(\Delta, \mathbf{A}, \mathbf{B})$
11:     $\mathbf{h}_0 = \mathbf{0}^{D \times N}$
12:     **for** $t = 1 \ldots T$ **in parallel (scan)**     # GPU hardware accelerated $\mathbf{y} \leftarrow \text{SSM}(\bar{A}, \bar{B}, C)(\mathbf{s})$.
13:         $\mathbf{h}_t = \bar{\mathbf{A}}_t \circ \mathbf{h}_{t-1} + \bar{\mathbf{B}}_t \mathbf{s}_t$     # Hadamard product for diagonal $\bar{\mathbf{A}}$.
14:         $\mathbf{y}_t = \mathbf{C}_t \mathbf{h}_t$
15:   **for** $t = 1 \ldots T$ **in parallel**
16:     $\mathbf{y}_t = \mathbf{y}_t * \mathtt{SiLU}(\mathbf{res_t})$
17:     $\mathbf{y}_t = \mathtt{Linear}_2(\mathbf{y}_t)$
18:   **return y**

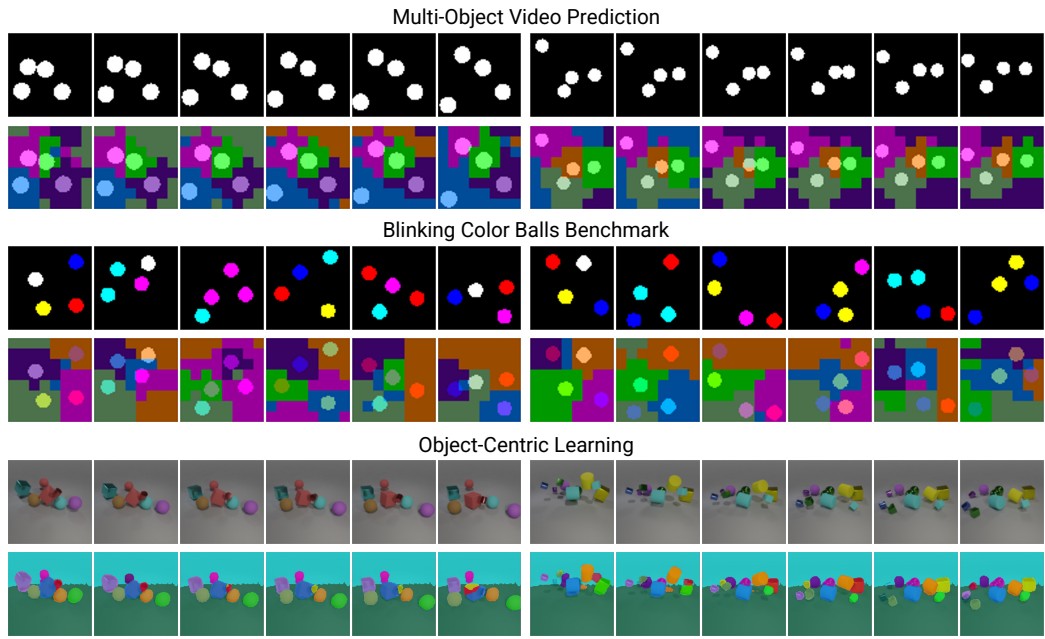

**Figure 12: Emerging Modularity in SlotSSMs.** Object-centric state representations naturally emerged to accommodate the underlying structure of the data.

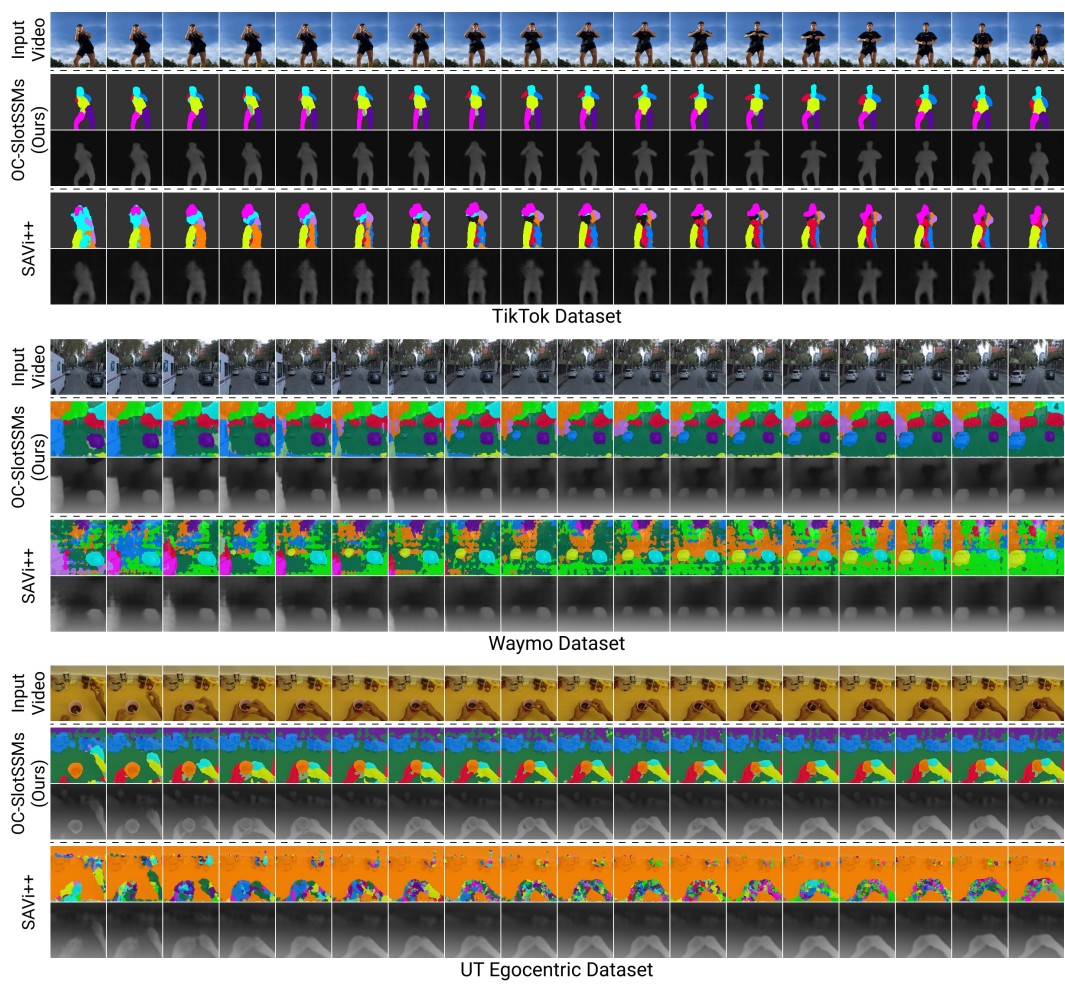

**Figure 13: Emergent Scene Decomposition from Depth Estimation Tasks.** Colors represent the ID of slots used for predicting each position. SlotSSM is capable of exploiting the inherent modular structure of real-world videos for efficient inference, without explicit segmentation supervision. For more examples please visit our project website.

