# OpenReview forum: "Slot State Space Models"
_NeurIPS.cc/2024/Conference — NeurIPS 2024 poster_

### Official Review · Reviewer_YJW6 · 2024-07-04

**Soundness:** 3
**Presentation:** 3
**Contribution:** 3
**Rating:** 7
**Confidence:** 4

**Summary:**

This paper introduces Slot State Space Models where instead of having a single monolithic state, the SSM state is divided into K different - each ideally representing a separate object in the scene. Each SSM state is evolved independently while interacting with each other through self attention. The complete architecture consists of Slot Encoder - to encoder images into slots, SlotSSM - to evolve the state of each SSM state, and Slot Mixer - to capture interactions between slots. Through experimentation they show the proposed architecture is useful for various long range reasoning and object-centric tasks.

**Strengths:**

The idea of incorporating modularity in State Space Models is interesting and combines the strenghts of SSMs (fast parallel training, constant memory inference) with that of modular architectures (factorization, compositionality, ood generalization).

The paper is well written and experimental results demonstrate the efficacy of the approach.

**Weaknesses:**

The paper has introduced a multi-layer architecture where the number of slots can vary per layer but this architecture has not been used in any of the experiments. Also if this architecture requires the user to specify the number of slots separately for each user, its utility seems limiting since it is hard to estimate this beforehand and would require abundant hyperparameter tuning.

The authors have stated that the interaction between slots is sparse. If the interaction is done using QKV self-attention, then each slot can interact with every other slot and hence the interaction should be dense. I don't understand how such an interaction can be sparse.

It would be nice to see an ablation which investigates the effect of number of slots. It seems that most experiments use 6 slots. Specifically, it would be interesting to see how the performance scales with increasing number of slots.

**Questions:**

- One of the issues with applying object-centric models to videos is the problem of temporal consistency - a given object may be represented by different slots into neighbouring timesteps. Did the authors face this issue in their video experiments and is there a way to address this in SSM based models which consider parallel training across timesteps? In RNN based method, this is addressed by initializing the slots of the future timestep by slots from the previous timestep.
- Would it be possible to apply top-k attention like used in RIMs in the proposed architecture? Do the authors see any benefit to doing that in this architecture.

**Limitations:**

The authors have discussed limitations.

---

> ### Author Rebuttal · Authors · 2024-08-05
>
> ## We sincerely appreciate for your valuable insights!
> > The paper has introduced a multi-layer architecture where the number of slots can vary per layer but this architecture has not been used in any of the experiments.
>
> In our current design, only our first layer changes the number of slots from # of input tokens to # of slots. We considered this model description to be more general since it can unify the formulation of the first layer and following layers. However, we agree that this could bring confusion to the readers, and we will revise this part in our revision.
> > requirement for selecting number of slots limits the model since it is hard to estimate beforehand and would require abundant hyperparameter tuning … It would be nice to see an ablation of number of slots.
>
> In general, using more slots leads to better performance in reasoning tasks. The number of slots presents a trade-off between performance and efficiency. In practice, we recommend starting with a redundant number of slots and gradually reducing it until performance drops. To investigate the effect of the number of slots, we conducted an ablation study using 1, 2, 4, 6, and 8 slots in the Long-Context Reasoning task with sequence lengths of 320 and 640. The results are provided in the following table:
>
> |Prediction MSE ( $\downarrow$ ) ( $\times 10^{-2}$ )|1 slot|2 slots|4 slots|6 slots|8 slots|
> |:-:|:-:|:-:|:-:|:-:|:-:|
> |Count Color: Length 320|4.379|1.170|1.009|0.965|**0.879**|
> |Count Color: Length 640|5.021|2.089|1.036|1.039|**0.857**|
>
> We see that the results do present an improving performance with increasing number of slots. We will include the ablation study in our revision.
> > If the interaction is done using QKV self-attention, then each slot can interact with every other slot and hence the interaction should be dense. I don't understand how such an interaction can be sparse.
>
> The term 'sparse' is used to contrast with the dense interaction in single state models, where all dimensions are continuously mixed across time and space through MLP or attention layers, making it difficult to isolate independent mechanisms into separate units. With SlotSSM, these representations are divided into independent slots that interact solely through self-attention layers after temporal updates, encouraging the learning of independent mechanisms. The term 'sparse' refers to the isolation of the dedicated interaction module and the independence in all other modules. We will clarify this part in the revised version.
> > Did the authors face temporal consistency issue in their video experiments and is there a way to address this in SSM based models which consider parallel training across timesteps? In RNN based method, this is addressed by initializing the slots of the future timestep by slots from the previous timestep.
> >
>
> We appreciate the reviewer's question. Firstly, please note that segmentation metrics like FG-ARI and mIoU in our object-centric learning experiment are evaluated temporally, and thus explicitly measure slot temporal consistency. The results demonstrate that our model achieves better consistency than the baseline in these tasks. We will explain how SlotSSM address the temporal consistency in the following:
>
> To allow for parallel training and fast inference, SlotSSM does not reuse previous slots for future initialization. Instead, temporal consistency is progressively promoted through deeper layers. In SlotSSM, the first slot encoder layer obtains frame-wise bottom-up information, enabling only partial temporal consistency from shared slot initializations across time. These slots are then refined by the SSM and Mixer layers, where the SSM injects temporal information and the mixer layer provides global awareness. Importantly, the mixer layer can route information between slots according to the updated temporal information to enable consistent capturing of moving objects/parts. These processes are repeated layer by layer to promote temporal consistency progressively. Optionally, we can also apply the slot encoder at deeper layers to further refine the input attention, as done in our OC-SlotSSM variant.
>
> In our rebuttal PDF, we include the Fig 3 `Emerging Modularity in SlotSSMs` to qualitatively show that this temporal consistency is achieved by both SlotSSM and OC-SlotSSM models across tasks. Additionally, the new real-world video results in the PDF demonstrate that temporal consistency is achieved on real-world videos as well. We will include a more detailed discussion in our revision.
> > Would it be possible to apply top-k attention like used in RIMs in the proposed architecture? Do the authors see any benefit to doing that in this architecture.
>
> We thank the reviewer for providing the insightful comment. We believe applying the top-k selection in SlotSSM would be possible and it would be an interesting topic. Potentially, it can improve the generalization and efficiency when number of slots is large. However, we note that directly applying RIM’s top-k selection might be suboptimal as it could pose optimization challenges. Our experience with SlotRNN vs RIM in Long-Context Reasoning tasks showed RIM to be significantly harder to optimize. We suspect the use of top-k selection, which restricts learning of the full set of slots at each iteration, may impact training stability. Therefore, we will emphasize the need to investigate approaches to stabilize training for top-k SlotSSM. Some of the possible approaches will be the following:
>
> - Initially training with full slots, followed by test-time top-k selection. The test-time top-k selection can be done by selecting the top-k slots with largest output norms as usually done in MoE literature [1].
> - Training with full slots initially, then fine-tuning the model on top-k selections post-training, essentially introducing an alignment stage for adapting to top-k selection.
>
> [1] A Closer Look into Mixture-of-Experts in Large Language Models, https://arxiv.org/pdf/2406.18219

---

> > ### Comment · Reviewer_YJW6 · 2024-08-07
> > **Thank you for rebuttal**
> >
> > I thank the author for their rebuttal. My questions have been addressed. I hope that the authors will add the clarifications they mentioned in the rebuttal to the paper. I am happy to raise my score.

---

### Official Review · Reviewer_Ae5c · 2024-07-12

**Soundness:** 4
**Presentation:** 2
**Contribution:** 3
**Rating:** 7
**Confidence:** 4

**Summary:**

This paper presents SlotSSMs, an extension of SSMs such that their states encourage information separation. This is in contrast to conventional SSMs whose state is monolithic. The authors evaluate slotSSMs in object-centric video understanding and video prediction tasks involving multiple objects and long-range temporal dependencies.

**Strengths:**

- As far as I know, this is the first work to propose the use of slots and object-centric modeling in SSMs.
- The paper is well structured and easy to read. The exhibition and definition of the different SlotSSM components is very clear (up to some small observations).
- The paper contains multiple experiments that illustrate the power of their proposed method. It also incorporates multiple ablation studies that illustrate the contribution of each of the proposed modules.

**Weaknesses:**

- My biggest concern lies in the evaluation of the method. For example, in Fig 4, the loss of SlotSSMs is relatively low and comparable to that of SlotTransformers. However, if we look at the predictions, these do not look good for any of the methods. I wonder if these methods and dataset are the right place on which SlotSSMs should be evaluated. Note that this is also the case in Fig 7, where all of the predictions are also quite bad. I would encourage the authors to think about settings on which SlotSSMs could really solve an open issue as this would increase the impact of the paper.
- Next, it seems that the authors evaluate other features than what is often evaluated in the literature. For example, in the MOVi paper, the authors evaluate object segmentation, for which SAVi gets 82% accuracy. I do not see this back in the paper, unless this is the first column of Fig 7 right, in which case the matrics are very low in comparison to the original paper. This, in combination with the reconstructions shown in the paper, I doubt that the Slot SSMs would be any better than SAVi. Also, the authors should consider including more recent methods in the comparison as well.
- In addition, it would be nice to see –as is often the case in slot papers– to check what the slots are actually learning. If this is not as easy for SSMs as for Transformers, then this should be stated as well in the limitations.
- I also have concerns regarding the reproducibility of the method. There are several experimental details missing which, in its current form, I would argue would make this paper irreproducible. Given that the metrics in this paper are also lower than what is shown in other papers, I consider this to be of vital importance. I would encourage the authors to add these details in the appendix.

**Questions:**

Aside from the previous weaknesses, there are some other aspects I would like to mention / clarify:

- In Eq. 8, are these Linear layers different per slot or are they the same everywhere?
- As far as I understand, in Eq.7 the “CLS tokens” are basically learnable embeddings. Is there a reason to call these CLS tokens instead? This might be confusing, as this implies that the input is always the CLS, which I think is not the case here.
- The authors go about how their method can use a different number of slots at each layer, but then go and use the same number of them at all layers. It would perhaps be better to introduce the slots refinement module as it is. In the current setting, when reading this it feels as if having a different number of slots will be required, which can be counterproductive for the impact of the paper (readers can think that the method is too complicated to use in practice).
- There are several typos / misspellings in the paper.
- As far as I know, the A matrix of Mamba is not input dependent. Could you confirm if you made it input dependent in Slot SSMs? Also, if you are using only Mamba, why not call the paper Slot Mamba?
- In many aspects, your proposed method is similar to the idea of using heads but for SSMs. Could you please comment on this? Also, I am aware that multiple works have introduced heads to SSM / long-conv models as well, e.g., in multi-headed Hyena [1].

[1] https://arxiv.org/abs/2310.18780

**Limitations:**

The authors clearly state a limitations section for the method. However, the limitations stated there obey more to a future work section and does not really discuss the limitations of the existing method. I would encourage the authors to revise this section and state the limitations of the submitted work.

### Conclusion

Whilst I acknowledge the novelty of this paper, I as of now have many concerns that I believe should be addressed before the paper is ready for publication. I therefore am unable to support acceptance. With that being said, I am happy to increase my score should these concerns be addressed.

---

> ### Author Rebuttal · Authors · 2024-08-05
>
> ## We sincerely appreciate your insightful feedback!
> > Fig 4 predictions do not look good
>
> We would like to rectify an error in Fi.g 4. We mistakenly used Single State SSM (Split)'s figures for "Ours,", and therefore the images does not represent our model's prediction accuracy. We have included an revised figure in rebuttal PDF Fig 2 left with enhanced visualization. The updated figures show more clearly that our model outperforms the baselines.
> > Fig 7 SAVi segmentation metrics are very low comparing to the original paper.
>
> Our experiment uses a different setting. The SAVi paper uses two additional and privileged training signals: (1) label conditioning, where the segmentation of each object in the first frame is given as input, and (2) an additional modality, where optical flow, instead of RGB reconstruction, is the prediction target, making the model less sensitive to high-frequency details in the RGB domain.
>
> In contrast, our Fig 7 (also see rebuttal PDF Fig 2 right) uses a fully unsupervised setting where learning is through video reconstruction. The original SAVi paper also provide such setting. A direct comparison can be made by comparing the **MOVi-A FG-ARI in our Fig 7** with the **MOVi FG-ARI of the SAVi (uncond. w/o flow) in Fig 3(a) of SAVi paper [1]**. The SAVi paper achieved ~62%, while we reproduce 63.98%. Thus, we are confidence that our implementation aligns with the original results. Also, we would like to emphasize that object-centric learning is just one possible application of the SlotSSM unlike SAVi. The proposed SlotSSM is a more general sequence modeling architecture.
> > Fig 7 Qualitative prediction looks bad.
>
> Fig 7 shows results for MOVi-B, a challenging dataset for all models due to its greater visual complexity compared to MOVi-A. We include MOVi-A results in rebuttal PDF Fig 2 right for comparison to more clearly show that our model outperforms SAVi.
>
> To improve robustness in visually complex datasets, recent approaches, such as SAVi++ [2], suggest using cross-modality signal in training. We have included new experiments in this setting, please refer to rebuttal PDF Fig 1 and our response to reviewer ERbD for more details.
> > Authors: Additional comments on evaluation metrics.
>
> We hope our above responses adequately addressed the concerns on evaluation metrics. Here, we clarify the motivations behind our evaluation. Our evaluation aims to highlight three key benefits of SlotSSMs:
>
> 1. Improved visual reasoning ability by using modular representations.
> 2. Improved long-context reasoning ability by employing SSM.
> 3. The emergent of object-centric representations from video reconstruction (OC-SlotSSMs).
>
> Our Multi-Object Video Prediction task verifies #1, Long-Range Reasoning task proposes a novel benchmark to highlights both #1 and #2, Object-Centric Learning task verifies #3, and 3D Visual Reasoning task showcases both #1 and #3.
> > It would be nice to check what the slots are actually learning.
>
> In SlotSSM, we can inspect the attention pattern in the decoder to see what each slot is learning. Results are shown in the rebuttal PDF Fig 3. Our findings suggest that SlotSSMs captures semantically meaningful components in different slots. Thank you for the suggestion. We will add this result in the revision.
> > There are several experimental details missing.
>
> We fully agree and will include detailed description of our design and experiments. Additionally, we are committed to releasing our code upon acceptance.
> > Linear layers per slot or the same? … are “CLS tokens” learnable tokens and why the naming?
>
> The linear layers are shared for all slots. The CLS tokens are learnable embeddings. Following the tradition of ViT, we use “CLS” to indicate that they are not observation tokens. We will clarify these in the revision.
> > slots number for different layers can be different in description but use same in practice … can be counterproductive for the impact
>
> Thank you for the thoughtful suggestions. The current description unifies the formulation of the first layer and the following layers. However, we agree that it could be complicated and will clarify this part in the revision.
> > the A matrix of Mamba is not input dependent?
>
> The A matrix is not input dependent. However, the discretized $\bar{A}$ is input dependent via the input-dependent $\Delta$, i.e., $\bar{A}=(I-\frac{\Delta}{2} \cdot A)^{-1}(I-\frac{\Delta}{2} \cdot A)$. We will clarify this in the revision.
> > why not call the paper Slot Mamba?
>
> Although we use Mamba as the base backbone, the proposed key contribution is not limited to Mamba's specific architecture. Instead, it is generally applicable to the parallel-trainable SSM architecture class. We use "SSM" to refer to the general SSM selection.
> > proposed method is similar to the idea of using heads but for SSMs.
>
> It might be seen that way. However, they are actually quite different. In multi-head methods, heads are continuously mixed through projection and MLP layers, making it difficult to learn independent mechanisms. Conversely, SlotSSM encourages independent learning by explicitly splitting representations into slots that interact independently with input features and across time, communicating only through self-attention layers after temporal propagation.
> > the limitations obey more to a future work, does not really discuss the limitations
>
> We appreciate the suggestions and will revise the limitations section. For example, we will include our finding that in the 3D visual reasoning task, SlotSSM achieves suboptimal performance without task-free pre-training. This suggests that for tasks with sparse training signals, the sequential nature of SlotSSM performs better with a pre-training phase to learn to effectively utilize information from all time steps.
> > There are several typos / misspellings in the paper.
>
> We will carefully review the manuscript and fix the grammatical issues.
>
> [1] https://arxiv.org/abs/2111.12594
>
> [2] https://arxiv.org/abs/2206.07764

---

> > ### Comment · Reviewer_Ae5c · 2024-08-12
> >
> > Dear authors,
> >
> > Thank you very much for your rebuttal. My concerns have been addressed. Do note that there are several things that must be clarified in the paper. Under the promise that the authors will include all of these corrections / complements in the final version of the paper, I am happy to raise my score.
> >
> > My score is now 7.

---

### Official Review · Reviewer_ERbD · 2024-07-12

**Soundness:** 4
**Presentation:** 3
**Contribution:** 3
**Rating:** 7
**Confidence:** 5

**Summary:**

This paper presents Slot State Space Models (SlotSSMs), a novel framework that integrates modular structures and inductive biases into State Space Models (SSMs) to improve sequence modeling. SlotSSMs maintain a collection of independent slot vectors and perform state transitions independently per slot, with sparse interactions managed through self-attention. This approach effectively captures the inherent modularity present in many real-world processes. Authors demonstrate substantial performance gains in object-centric video understanding and video prediction tasks, highlighting the importance of modularity in sequence modeling. Additionally, the authors introduce Object-Centric SlotSSMs (OC-SlotSSMs), which leverage inverted attention to further enhance the discovery of modular structures. Extensive experiments across multiple tasks, including multi-object video prediction, long-context reasoning, unsupervised object-centric learning, and 3D visual reasoning, validate the effectiveness and versatility of the proposed models. The paper also includes a detailed analysis of the emerging modularity in SlotSSMs, showcasing their ability to naturally discover and exploit the underlying structure of the data.

**Strengths:**

- This approach is original in its design, utilizing independent slot vectors with sparse interactions managed through self-attention.
- A thorough evaluation of the proposed models across multiple challenging tasks, including multi-object video prediction, long-context reasoning, unsupervised object-centric learning, and 3D visual reasoning. The experiments are well-designed and provide robust evidence of the models' effectiveness.
- The authors provide a detailed analysis of the emerging modularity in SlotSSMs, offering valuable insights into how the models learn to capture the underlying structure of the data, with visual aids.
- The success of SlotSSMs in capturing modular structures suggests promising directions for further research in modular and object-centric sequence modeling. This could lead to the development of even more advanced architectures capable of handling complex, real-world data.

**Weaknesses:**

- While the potential for applying SlotSSMs to other modalities (e.g., text, audio) is mentioned, the paper does not provide experiments or theoretical analyses in these areas. Including preliminary results or theoretical discussions on applying SlotSSMs to different modalities would strengthen the paper and demonstrate broader applicability.
- In the related work section, mention also another interesting work utilizing SSMs for Neuromorphic Cameras: "State Space Models for Event Cameras". Nikola Zubić, Mathias Gehrig, Davide Scaramuzza.

**Questions:**

1. Given the current computational constraints mentioned in the paper, what are the potential strategies for scaling SlotSSMs to handle larger datasets and model sizes? Are there specific optimizations or approximations that you are considering to improve scalability?
2. Have you conducted any preliminary experiments or theoretical analyses on applying SlotSSMs to other data modalities, such as text or audio? What challenges do you anticipate in these domains, and what benefits might SlotSSMs bring?
3. How do you anticipate SlotSSMs would perform on datasets with higher visual complexity? Have you identified any specific challenges or limitations in such scenarios, and what strategies might you employ to address them?

**Limitations:**

Already discussed by the authors.

---

> ### Author Rebuttal · Authors · 2024-08-05
>
> ## We sincerely thank you for your positive recommendation and thoughtful comments!
>
> > … the paper does not provide experiments or theoretical analyses in other modalities (e.g., text, audio). …  What challenges do you anticipate in these domains, and what benefits might SlotSSMs bring?
> >
>
> Thank you for the insightful suggestions. We agree that additional experiments on other modality would make the paper stronger. We’re planning to explore this aspect in our next project. We will include additional discussion in the following about modality-agnostic generality of SlotSSM in our revision:
>
> - **Audio modality.** A key feature of SlotSSM is its ability to leverage the slot representations to exploit the modular structure of input data for downstream tasks. This is especially relevant for videos, which are inherently modular, yet require the model to uncover this structure. We anticipate this principle extending to audio inputs, where multiple acoustic sources may exist, and the model must inherently decompose these sources to understand the context. In such scenarios, slots can represent the decomposed acoustic sources.
> - **Text modality.** Another important aspect of SlotSSM is the introduction of additional memory units, the slots, at each time step. Unlike conventional SSM-based language models that use a single state for each input token, slots can store additional pertinent information extracted from past data. Therefore, applying SlotSSM to process language tokens could potentially enhance the capabilities of existing language models, improving performance on tasks such as long-context reasoning.
>
> > In the related work section, mention also another interesting work Neuromorphic Cameras: "State Space Models for Event Cameras". Nikola Zubić, Mathias Gehrig, Davide Scaramuzza.
> >
>
> Thank you for suggesting this. We will mention it in the related work.
>
> > … what are the potential strategies for scaling SlotSSMs to handle larger datasets and model sizes? … How do you anticipate SlotSSMs would perform on datasets with higher visual complexity?
> >
>
> We appreciate this important question. We first want to highlight that our experiments have already demonstrated the significantly superior efficiency of the SlotSSM design compared to both transformer and RNN baselines. To further improve the efficiency in large scale datasets and model size, we can consider adopting the following strategies:
>
> 1. Utilizing pre-trained visual encoders for visual encoding and apply SlotSSM on top. This improves both representation quality and computation cost.
> 2. For object-centric learning, utilize a stronger image decoder, such as autoregressive transformer decoders or diffusion decoders. This improves reconstruction capability to handle visually more complex data.
> 3. Reducing the number of tokens for deeper layers to capture increasingly abstract information. This reduces the memory consumption for larger model size.
>
> Moreover, recent methodologies like SAVi++ [1] suggest leveraging cross-modality signals to facilitate object-centric learning on real-world videos without extensive scaling of training. Inspired by this, we conducted an additional experiment where the model received RGB video inputs and predicted the depth of each frame. We provide qualitative results in rebuttal PDF Fig 1.
>
> We compared OC-SlotSSM with SAVi++ on three real-world datasets: UT Egocentric Videos, Waymo Autonomous Driving Videos, and TikTok Dancing Videos. Qualitative visualizations on the TikTok Dataset are provided in the rebuttal PDF Fig 1. Our OC-SlotSSM model demonstrated its ability to exploit the inherent modular structure of the videos to accomplish the task, with unsupervised scene decomposition emerging during training. Below, we provide a quantitative comparison.
>
> |      Prediction MSE ( $\downarrow$ ) ( $\times 10^{-3}$ )          | UT Egocentric       | Waymo Autonomous Driving     | TikTok
> |:----:|:-----:|:----:|:----:|
> | SAVi $++$          | 0.5885 | 0.804 | 1.412 |
> | OC-SlotSSM (Ours) | **0.4640** | **0.653** | **1.180** |
>
>
> We see that our OC-SlotSSM consistently outperformed the SAVi++ baseline across datasets, showcasing its superior video modeling capabilities for this task. During the experiment, we also noted that OC-SlotSSM maintained better training stability, particularly with longer sequences. In our revised manuscript, we will include more qualitative results on the emergent scene decomposition and detailed discussions comparing the performance and scalability of OC-SlotSSM and SAVi++.
>
> [1] SAVi++: Towards End-to-End Object-Centric Learning from Real-World Videos, https://arxiv.org/abs/2206.07764

---

> > ### Comment · Reviewer_ERbD · 2024-08-08
> >
> > 1. Authors briefly explained how the future work for audio and text modalities would look like.
> > 2. Authors will cite "State Space Models for Event Cameras" in the Related work section.
> > 3. Authors addressed my question on larger datasets and model sizes, also the higher visual complexity question.
> > 4. They did a nice additional experiment.
> >
> > Therefore, all of my concerns were addressed and I finalize my rating as:
> >
> > 7: Accept

---

### Author Rebuttal · Authors · 2024-08-05

We thank all reviewers for their insightful and positive feedback! We are encouraged that they find our work **novel** (ERbD, Ae5c), **interesting** (YJW6), and **offers valuable insights** (ERbD). They also highlighted its **potential to facilitate future research** and **lead to more advanced architectures** (ERbD). We are pleased that they recognized our empirical evaluation as
**thoroughly conducted** (ERbD), **demonstrating models' effectiveness** (ERbD, YJW6), and our paper **well-written** and **easy to follow** (Ae5c, YJW6).

We provide additional experiment results in our uploaded pdf. These experiments are done to respond to reviewer **ERbD’s question about model performance on visual complex dataset**, **Ae5c’s question about prediction quality and suggestion to check what the slots are actually learning**, and **YJW6’s question about slots’ temporal consistency**. A detailed overview of the PDF is the following:

1. **In Figure 1**, we conduct additional experiments for depth estimation on three real-world video datasets to showcase SlotSSM’s ability to handle visually complex real-world videos. In the PDF, we use TikTok dataset as an example to show the emergent unsupervised scene decompositions and depth prediction results, demonstrating that SlotSSM is able to utilize the modular representations to discover and exploit the latent structure of the input to complete the task. Quantitative results is shown in the rebuttal.
2. **In Figure 2**, we provide revised figures for video prediction and object-centric learning experiments to better showcase SlotSSM’s superior performance.
3. **In Figure 3**, we visualize the attention pattern in the decoder to investigate what each slot is learning and how they contribute to the final prediction. Results demonstrate the emergent modularity in slot representations as well as their temporal consistency.

We will respond to each reviewer’s concerns and questions separately below.

---

### Decision · Program_Chairs · 2024-09-25

**Decision:**

Accept (poster)

**Comment:**

Overall, after reading the reviews and the rebuttal, I think the paper is in great shape and should be accepted. Echoing what the reviewers said, I think the method is sound, and the paper does a great job at ablating and understanding how the proposed slotSSM structure works, showing good results.
Please consider all the feedback received, and incorporate in the camera ready all the changes/additional results mentioned in the rebuttal.